# Enhancing the performance of Magnets photosensors

Armin Baumschlager [1,4] ✉, Yanik Weber [1,2,4], David Cánovas [3,4], Sara Dionisi[1] & Mustafa Khammash [1] ✉

Photosensory protein domains, derived from nature, are foundational for optogenetic protein engineering. Tailoring their properties enables their full exploitation for optogenetic regulation in basic research and applied bioengineering applications. Here, we present a simple, yet powerful strategy based on random mutagenesis coupled to high-throughput screening that allowed altering the most fundamental properties of the widely used nMag/pMag photodimerization system: its light sensitivity and activation. Variants were characterized in vivo in bacteria by flow cytometry and during the entire growth curve by spectrofluorometry. We identify mutations that either increase or decrease the light sensitivity at sub-saturating light intensities, while also improving the light activation and dark-to-light fold change. Notably, light sensitivity and activation levels could be changed independently. In addition, we demonstrated that the shapes of the dose-response curves can be finely tuned. This broadens the applicability of the Magnets photosensors for optogenetic regulation strategies.

In bioengineering, light has emerged as a versatile input to steer and control various biological processes[1,2]. It is crucial for photosynthetic processes or protection against the harmful effects of electromagnetic radiation, such as photooxidative cellular damage. Organisms have evolved a variety of photosensitive proteins called "photoreceptors" or "photosensors", which enable them to respond to light stimuli. Depending on their function, these domains can register different wavelengths and intensities of light. Light is usually absorbed through an organic chromophore, which induces structural changes within the apoprotein[1,2]. These structural changes revert to the ground or "dark" state of the photoreceptor in a thermally driven reaction.

Optogenetics employs such photosensors or photosensory domains for the engineering of genetically encoded light-sensitive proteins[1–3]. Since the development of the first genetically engineered light-sensitive proteins[4], light has been used in different optogenetic approaches to interfere with and control a variety of cellular functions[3,5,6]. The use of light as input is particularly attractive because it enables precise spatial and temporal applications[7,8], both with a higher resolution when compared to chemical induction. Furthermore, in non-photosensitive organisms, such as *Escherichia coli*, light is an orthogonal input that does not require uptake and/or conversion as with small molecule inducers[9–11]. Thus, the use of light allows for minimally invasive spatiotemporal control.

Photosensitive domains are the key components of optogenetics. In general, optogenetic proteins comprise a photosensory domain and an actuating module[3,5]. For example, many cellular sensors, such as kinases, initiate signaling and cellular responses through oligomerization[12,13]. Such proximity-based regulation is also the basis of various natural light-sensing modules in which the interaction, and therefore the distance of the interaction partners, is controlled through light in a mechanism that leads to either homodimerization or oligomerization of the same photoregulator or heterodimerization and oligomerization of two or more different photoregulators. Such proximity control can be used to assemble inactive subcomponents into an active protein, or for the recruitment of an active protein to a specific location of action. Light-controlled dimerization domains have

[1]Department of Biosystems Science and Engineering (D-BSSE), ETH Zürich, Basel, Switzerland. [2]Institute of Pharmacology and Toxicology, University of Zurich, Zurich, Switzerland. [3]Department of Genetics, Faculty of Biology, University of Seville, Seville, Spain. [4]These authors contributed equally: Armin Baumschlager, Yanik Weber, David Cánovas. ✉e-mail: armin.baumschlager@me.com; mustafa.khammash@bsse.ethz.ch

been used to implement optogenetic control usually through homo-dimerization of e.g., receptors[14–20] or heterodimerization for split proteins[8,21–23], as well as subcellular localization[4,24,25].

A widely used class of photosensors is the light, oxygen, and voltage (LOV) domain, which has features that are often especially attractive for optogenetic protein (Opto-protein) engineering, such as a small domain size and tunable kinetic properties. LOV domains employ flavin mononucleotide (FMN) and flavin adenine dinucleo-tide (FAD), which are present in most organisms, as chromophores. One of the most used photosensory protein heterodimerization pairs is the "Magnets"[26], which is derived from the homodimerization protein VIVID (VVD). The LOV domain VVD, from the filamentous fungus *Neurospora crassa*[27], has been used in numerous optogenetic designs ([14,19,28–39]). Blue light induces a conformational change in the LOV domain, which initiates the homodimerization of two VVD domains and dimer stabilization[27,40,41]. Based on a truncated version of this photosensory domain[27], the Magnets heterodimerization system was created through rational protein engineering[26]. For this, positively and negatively charged amino acids were introduced into the VVD dimer interface to create complementary pMag and nMag domains[26,42]. We and others used these Magnets domains for the reconstitution of split proteins[8,29–32] due to their small size (150 amino acids), favorable structure in which N- and C-termini of the two domains come in proximity in the dimeric state, and low dark-state activity as well as reversion rate tunability[8]. In prior work, mutations in the Magnets domains and VVD were identified using rational structure-guided protein engineering. These approaches enabled changing their kinetic properties[20,26,27,43,44] or optimizing function-ality in mammalian cells[45].

The Opto-T7RNAP is a light-inducible transcription system that incorporates the VVD-based Magnets photoregulators into the het-erologous T7 RNA polymerase (Fig. 1A). It consists of two split frag-ments of the T7RNAP, which are fused to the light-inducible heterodimerizing Magnets domains. nMagHigh1 is fused to the C-terminus of the T7RNAP N-terminus, and pMag is fused to the N-terminus of the C-terminal T7RNAP fragment. Light induces a conformational change in the Magnets domains, which leads to the binding of the two complementary nMagHigh1 and pMag domains. The resulting spatial proximity of the T7RNAP fragments restores the enzyme's function, enabling transcription of genes from T7 pro-moters. The screening system consists of two plasmids, one con-taining the two Opto-T7RNAP genes under the control of arabinose-inducible promoters, and a reporter plasmid containing the red fluorescent protein mCherry expressed from the T7 promoter. The gene expression system shows high activation with blue light induction and simultaneously low residual expression in the dark[8]. This ratio between the expression level in blue light and in the dark is called the light-induced fold change. A crucial aspect for the use of this system during a screen is that it offers a high dynamic range, which enables screening at sub-saturating induction levels (Fig. 1B, C). We characterized this transcription system via the expression of the red fluorescent protein mCherry (Fig. 1A). We investigated the effect of different ratios of the two Opto-T7RNAP split parts and observed that adjusting the relative ratios of the two components led to different levels of reporter gene expression. However, the dose-response curves, and thus the light sensitivity of the Opto-T7RNAPs, were not changed due to the altered regulator expression levels. (compare Fig. 2B, C and Fig. S11 "Light dose-response curve is not changed by different domain expression levels" of Baumschlager et al. [8]). Thus, changes in light sensitivity cannot be achieved by adjusting the ratios of the Opto-T7RNAP parts.

Furthermore, we investigated how the expression levels of both Opto-T7RNAP parts influence the properties of the system. Since both components are expressed from the same arabinose-inducible pro-moter, this was achieved by varying the concentration of the arabinose inducer. Despite an increased Opto-T7RNAP protein expression, we only observed moderate (4-fold) changes in the resulting reporter expression comparing both dark and light-induced conditions sepa-rately (Fig. 4 of the publication Baumschlager et al. [8]). In addition, we observed that only relatively high arabinose induction (more than 0.05% arabinose) leads to significant changes in the resulting output (same figure). These observations suggest that the inherent light-sensing properties of this system cannot be addressed simply by changing the levels of the regulatory proteins or by altering the relative concentrations of the individual split components.

Further, when incorporating the previously identified photocycle mutations I85V (pMagFast1) and I74V, I85V (pMagFast2), we could only observe a change in the dose-response curve when pMagFast2 was incorporated. (Fig. 4 of Baumschlager et al. [8]) For both variants, we observed large changes in the reporter expression level at saturating conditions (Fig. 4, Fig. S13 of Baumschlager et al. [8]). Further, only the variant containing pMagFast2 exhibited a change in the light dose-response curve (Fig. 4 and Fig. S11 of Baumschlager et al. [8]), and only the other variant pMagFast1 showed faster off-switching kinetics. Importantly, this suggests that different properties of photosensitive proteins, such as light sensitivity, lit-state lifetime, or light-activation, can be tuned independently. These results prompted us to investigate these hypotheses further, leading us to develop a method that enables independent tuning of different optogenetic parameters.

Here, we envisioned that the direct phenotypic output of our transcription system in *E. coli* enables high-throughput screening for the desired photoregulator properties. These properties range from "light sensitivity" alone to "light activation" alone, or a combination of both, extending previous protein engineering efforts. Light sen-sitivity is used in this work to describe changes in light intensity that result in half-maximal activation. In contrast, light activation describes changes in the resulting reporter protein expression level in this manuscript. Fold change or light-induced fold change is defined as the ratio between the reporter expression in light and the expression in dark.

Specifically, in this work, we aimed to alter the most fundamental properties of this photosensory domain: its light sensitivity and its light activation. Highly light-sensitive photosensors, for example, bear the advantage that lower-intensity light can be applied to achieve similar outputs. This reduces potential phototoxic effects (e.g., mam-malian cell lines[46]), aids in light delivery into denser cell cultures[5,47], and minimizes heat development by the light application. The same can be achieved by increasing the activity of the optogenetic regulator (light activation), even if the shape of the dose-response curve remains unchanged. In other scenarios, lower light sensitivity might be pre-ferred, e.g., if a regulator shall not be activated by ambient light, for expression of (potentially) toxic proteins, or for multiplexing when combined with other, more sensitive, light controllers.

To allow for screening of Magnets variants with increased or decreased light sensitivities and light activation, we used the Opto-T7RNAP*(563), which directly links the activity of the photosensitive domain to a detectable fluorescence output, thus creating a genotype-phenotype linkage that allows for screening of variants with altered photosensitivity. Through a single round of mutagenesis, we identified several mutations in both the nMagHigh1 and the pMag domains, some of which led to dramatically increased light sensitivities while main-taining or even improving the dark-to-light fold-change or changing the light activation (reporter expression levels). While the mutations identified here will expand the applicability of these photosensory domains and enable a wider range of applications, we anticipate that our approach can also be used to identify mutations that change other properties, such as dark-state reversion rate or enhancement of fold change, which will further expand the applicability and toolset of VVD-based photoregulators in particular, and optogenetic approaches in general.

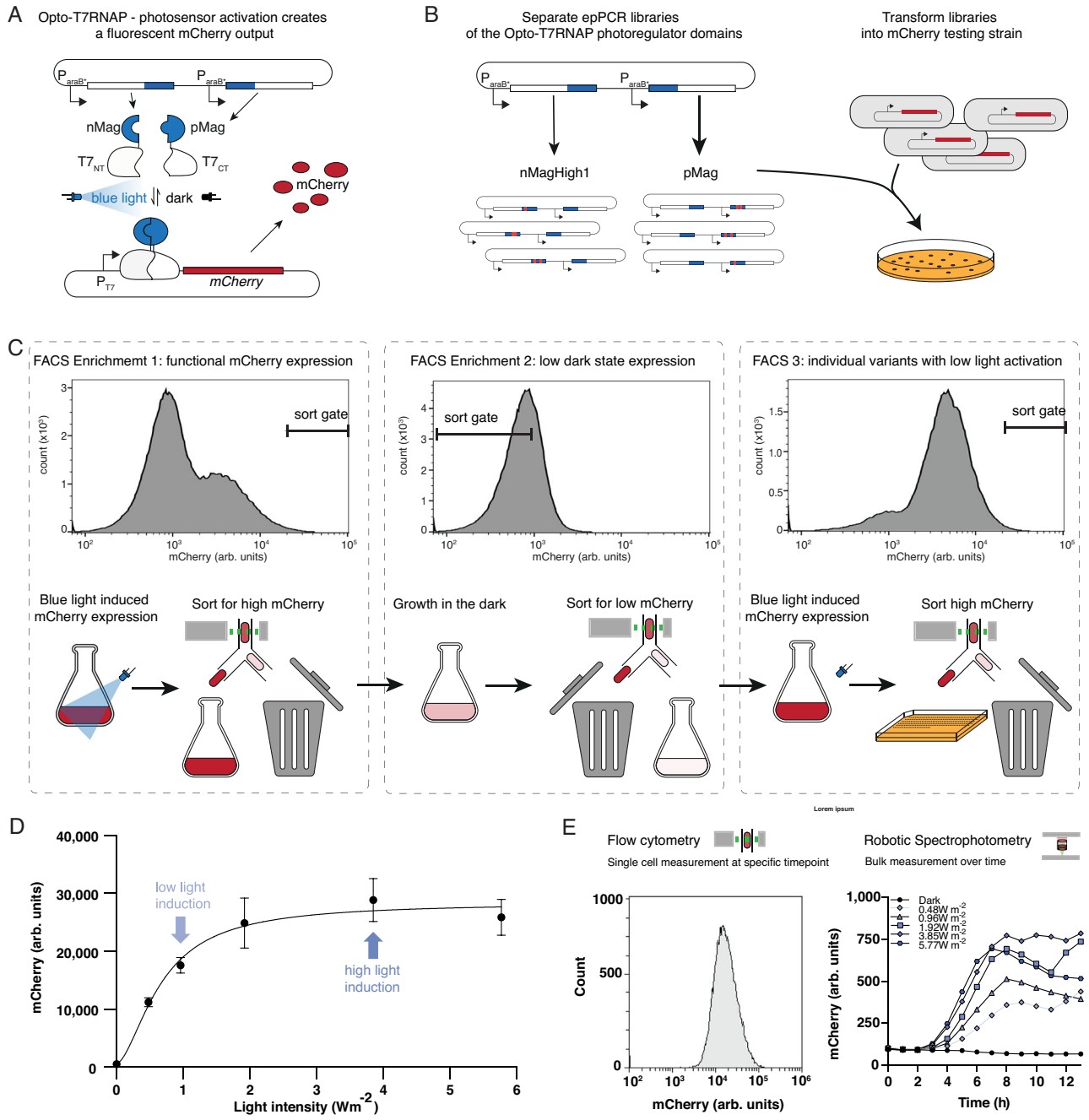

**Fig. 1 | Vivid-based Magnet photosensor engineering using error-prone PCR and the optogenetic transcription system "Opto-T7RNAP" via Fluorescence Activated Cell Sorting (FACS).** **A** Blue-light activatable T7 RNA Polymerase (T7RNAP) Opto-T7RNAP[8] and mode of action. mCherry fluorescence serves as a measurable output for Magnet photoreceptor heterodimerization. **B** Individual error-prone PCR libraries of nMagHigh1 and pMag photoregulators were constructed and transformed into an *E. coli* strain that contained the mCherry reporter plasmid. **C** Example FACS strategy to screen for variants with increased light sensitivity, meaning a higher mCherry expression level at non-saturating light-induction conditions. Shown are the histograms of the mCherry expression profile of a nMagHigh1 library with low-intensity light induction of the original mutagenesis library left, this enriched library regrown without light-induction in the middle for the second enrichment, and the second enriched library again induced with light-intensity light induction. All histograms further show exemplary sort gates which select for high mCherry expression with light-induction and low mCherry

expression in the dark. **D** Dose-response curve of the original Opto-T7RNAP*(563) of mCherry expression level in response to blue light (465 nm) of different light intensities. To be able to screen for higher light-sensitive photoregulators, 0.96 W m⁻² light intensity was used for sub-saturating induction, and 3.85 W m⁻² for saturating light induction of the libraries. For characterization of identified variants, we used both non-saturating (0.96 W m⁻²) and saturating (3.85 W m⁻²) light induction. The diagram shows mean mCherry expression values and standard deviation (mean values +/- SD) of six (*n* = 6) biological replicates measured after 5 h incubation time in all cases. **E** Characterization of variants was performed on the single-cell level through flow cytometry (left) and in batch culture over time through spectrophotometry (right) exemplarily shown with the wild-type Opto-T7RNAP*(563) regulator. Single-cell mCherry expression profile is shown after 5 h saturating light induction at 37 °C. Spectrophotometric measurements in batch culture over 13 h at 37 °C. The diagram shows mean mCherry expression values of at least eight (*n* = 8) biological replicates.

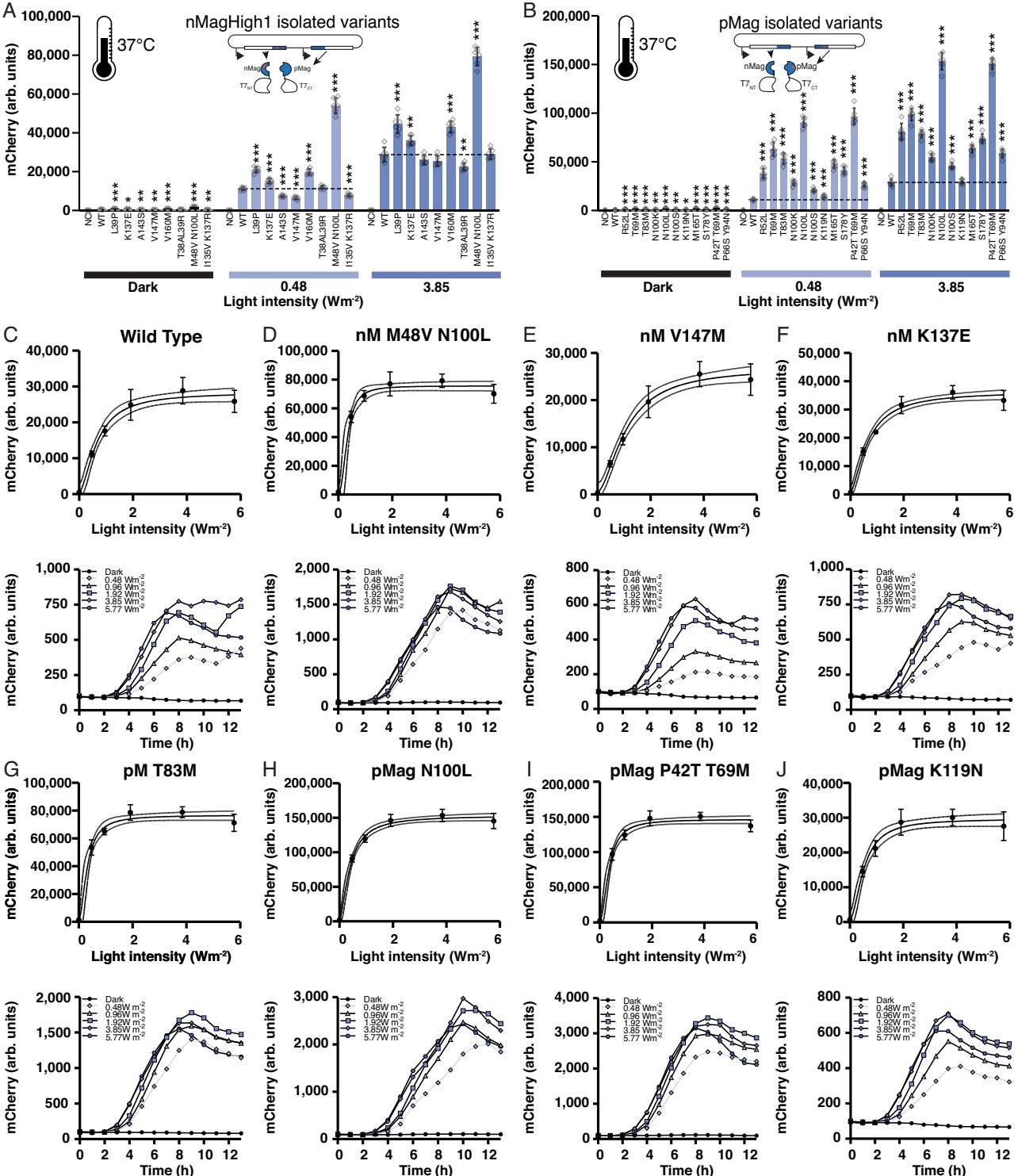

**Fig. 2 | Characterization of identified Magnets variants.** Mutations were found in nMagHigh1 (**A**) and pMag (**B**). Characterization of variants was performed through flow cytometry in comparison to the wild-type Opto-T7RNAP*(563) regulator and a negative control (NC) containing the mCherry expression plasmid and a second empty plasmid without the optogenetic regulator. mCherry expression values were acquired after 5 h incubation at the indicated light intensity (465-nm light) and 37 °C. Shown are the mean fluorescence values and standard deviation (mean values +/- SD) as well as individual data points of six (n = 6) biological replicates. Statistical significance was determined by Student's t-test comparing each variant to the wild-type strain (* p < 0.05, ** p < 0.01, *** p < 0.001). In addition, the mCherry expression was significantly higher in light than in the dark (t-test, p < 0.001) of all variants and wild type. Fluorescence measurement over time through spectrophotometry at the indicated light intensity and 37 °C of the wild type (**C**) and selected variants M48V N100L (**D**), V147M (**E**), K137E (**F**) of nMagHigh1 and T83M (**G**), N100L (**H**), P42T T69M (**I**) and K119N (**J**) of pMag. C–J Upper panels: shown are the mean fluorescence values and standard deviation (mean values +/- SD) of six (n = 6) biological replicates. (C-J) Lower panels: Shown are the mean fluorescence values of (n = 8) biological replicates for samples wild type at dark, 0.96 Wm⁻², 5.77 Wm⁻², pMag T83M at 1.92 Wm⁻², and pMag N100L at 5.77 Wm⁻², and (n = 9) biological replicates for all other conditions and variants.

## Results

### Library and experimental designs

We used error-prone PCR (epPCR) to diversify both photosensory domains (nMagHigh1 and pMag) of our Opto-T7RNAP*(563) individually (Fig. 1B). Both Magnets domains are based on a 36-residue N-terminal truncated VVD photoregulator[27], and differ in several amino acids that were rationally introduced to enable heterodimerization[26]. nMagHigh1 is C-terminally fused to the N-terminal part of the T7RNAP, while pMag is N-terminally fused to the C-terminal part of the T7RNAP (Fig. 1A). Although both photosensory domains are derived from VVD, we chose to create libraries for each individually (Fig. 1A, B), as they differ in amino acid positions I52 and M55 that confer complementary binding, as well as M135 and M165 for nMagHigh1[26]. This leaves the possibility for different compensatory mutations for each domain, as well as accounts for their fusion to either the N- or C-terminal part of the T7RNAP[8]. These libraries were then transformed separately into *E. coli* strain AB360[8], which contained a plasmid for expression of mCherry under T7RNAP control (Fig. 1B). We used two different ribosome binding sites (RBS) strengths, first the published RBS from pAB50[8], as well a version denoted "pAB50-11k" in which we weakened the TIR using the RBS calculator 2.0[48] to 15% of the original RBS (Supplementary Fig. 1). The use of these two different expression strengths in separate libraries should allow one to find higher light activation variants while limiting the effect of metabolic burden induced through mutations that cause higher mCherry expression levels which might obscure a screening.

We chose photosensitivity and photoactivation as the target properties for tuning for four main reasons: First, as previously mentioned, the ability to sense light effectively is the defining feature of a photoregulator, and engineering enzymes to directly improve their activities on their natural function is often unsuccessful and generally considered challenging[49]. Second, a potential drawback of optogenetic methods for the control of cells is light-induced cellular damage caused by high-intensity light and/or long illumination durations. While this effect might be less pronounced in bacteria like *E. coli*, it can be problematic in higher organisms, such as mammalian cell lines. Third, depending on the light application setting, a high light intensity can produce heat, which in turn might interfere with experiments or applications. Fourth, a set of photoregulators with different sensitivities to the same wavelength will enable the multiplexing of expression levels of different proteins, as well as the activation of genetic circuitry through variation in light intensity.

### Library induction and screening strategy

We applied a multi-step Fluorescence-Activated Cell Sorting (FACS) for the screening and selection of variant libraries with altered light sensitivities/activation (Fig. 1C). For this, we used different sequences of induction conditions: No light induction (further denoted "dark"), in which variants with mCherry expression comparable to the non-induced original Opto-T7RNAP*(563) were sorted for, or with light intensities of 0.96 and 3.85 W m$^{-2}$ (further denoted as low or high light induction respectively), which correspond to intermediate or saturating light induction (Fig. 1D). The example in Fig. 1C shows the screening approach for variants with increased light sensitivity. For the first enrichment (Fig. 1C left) nMagHigh1 or pMag libraries were induced with non-saturating light for 4 h and then sorted for high mCherry expression, while cells with low or no mCherry fluorescence were discarded. Most variants showed reduced-to-non-functional mCherry expression, like the negative control (Supplementary Fig. 2, left), indicating that most mutations inactivated the function of the photosensor, as expected through random mutagenesis. This removed the bulk of the library containing mutations that inactivated or reduced the light sensitivity of the photosensory domain. The enriched libraries were re-grown in the dark and then sorted for low mCherry expression for the second enrichment (Fig. 1C middle). The

second step served to select variants that still showed a low dark state and removed constitutively active variants. These enriched libraries were then again induced for 4 h with low-intensity 465-nm light blue light and single cells spotted on Omnitrays with LB-agar to isolate individual variants (Fig. 1C right), which were subsequently regrown in M9 medium and screened. Individual mutations were identified by Sanger sequencing and recloned into the original plasmid. For final characterization, cultures were grown and measured both through flow cytometry and monitored periodically through spectro-photometry until the stationary growth phase (Fig. 1E) as described in the Methods section.

### Identified nMagHigh1 and pMag variants

We compared the mCherry expression of the identified variants to the expression of the original wild-type Opto-T7RNAP*(563)[8] and a negative control, containing only the mCherry reporter plasmid and an empty backbone with the same antibiotic resistance as the plasmid harboring the Opto-T7RNAP. For this, we used two experimental approaches: first, single-cell analysis by flow cytometry at a single time point; and second, spectrophotometry over time.

Single-cell characterization of variants through flow cytometry was performed after 5 h expression in log growth phase. This data was used to extract changes in basal dark (denoted *b*) and maximal light-induced (denoted *t*) mCherry expression, which are important for assessing changes in Opto-T7RNAP light activation. For the changes in light sensitivities, we compared the light intensities that lead to half-maximal gene expression (denoted *I50*), which were obtained from fits of the light-induced dose-response of mCherry expression to a Hill-type equation, as described in the Materials and Methods section. Second, we also characterized the variants at the population level over time using spectrophotometry. The data were analyzed both by hier-archical clustering and by fitting the mCherry expression data at a single time point to a Hill-type equation to obtain the light-induced dose-response curve parameters.

After just one round of mutagenesis, we selected 19 different variants displaying a variety of interesting properties (Fig. 2 at 37 °C, Fig. 3A, B at 30 °C, and Fig. 3C, D at 40 °C; shown in more detail in Supplementary Fig. 3 and Supplementary Fig. 4). From these variants, five single and three double mutations were found in nMagHigh1, and nine single and two double mutations were located in pMag (Supplementary Table 1). All calculated properties for *b*, *t*, and *I50* as well as percentage changes compared to the WT are summarized in Supplementary Table 2-Supplementary Table 12 with the underlying data shown in Supplementary Fig. 5-Supplementary Fig. 20. Apart from these variants, we also isolated a double mutant in the pMag library (M135I, M165I), which was previously described as pMagHigh1[26], and showed a higher mCherry expression level compared to pMag in the screening, thereby confirming the ability of this setup and approach to identify variants with improved properties for a given experimental setup.

Reproducible robotic culture inoculation, followed by character-ization of the variants at the population level through spectro-photometry over time, enabled us to compare population growth curves. Neither the negative nor the wild-type controls showed apparent changes in their growth curves due to the light illumination and the mCherry reporter expression. This was also the case for most nMagHigh1 and pMag variants, with the exceptions of pMag P42T T69M, pMag R52L, pMag N100L, and pMag P66S Y94N at 30 °C. These variants showed a clearly decreased hill slope in the growth curve with increasing light intensity (Hill slopes shifted from 0.3013 to 0.1941 for pMag P42T T69M, from 0.3127 to 0.1858 for pMag R52L, from 0.2736 to 0.2012 for pMag N100L, from 0.3123 to 0.2174 for pMag P66S Y94N in the dark and with 5.77 Wm$^{-2}$ light induction respectively; see Sup-plementary Table 13 and Supplementary Table 14) compared to both the wild type (from 0.302 to 0.2631 Hill slope in the dark and with

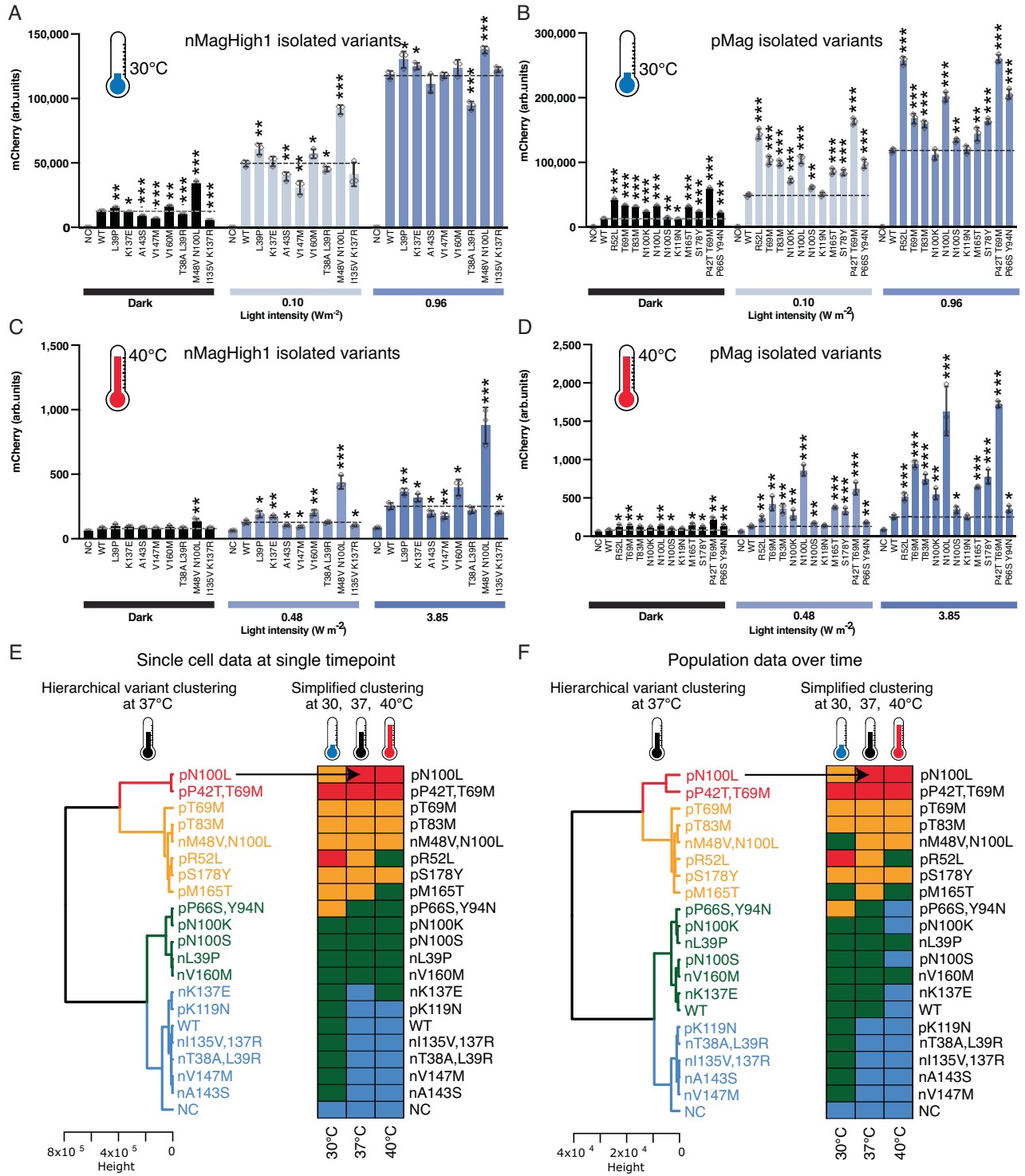

**Fig. 3 | Temperature behavior and hierarchical clustering of Magnets variants.** Characterization of identified photoregulator variants found in nMagHigh1(**A**, **C**) and pMag (**B**,**D**) at 30 °C (**A**,**B**) and 40 °C (**C**,**D**). Characterization of variants was performed through flow cytometry in comparison to the wild-type Opto-T7RNAP*(563) regulator and a negative control (NC) containing the mCherry expression plasmid and a second empty plasmid that does not contain the opto-genetic regulator. mCherry expression values were acquired after 5 h incubation at the indicated light intensity (465 nm light) and temperature. Shown are the mean fluorescence values and standard deviation (mean values +/- SD) as well as indivi-dual data points of three ($n = 3$) independent biological replicates. Statistical sig-nificance was determined by Student's *t*-test comparing each variant to the wild-type strain (* $p < 0.05$, ** $p < 0.01$, *** $p < 0.001$). In addition, the mCherry expression was significantly higher in light than in the dark (*t*-test, $p < 0.05$) of all variants and wild type. **E,F** Hierarchical clustering using the Ward method and Euclidean distances of flow cytometry data at one single time point (E) and spec-trophotometry data acquired every hour until stationary phase (24 h at 30 °C, 14 h at 37 °C, 17 h at 40 °C) (F). Variants were clustered according to the expression levels into very high (red), high (orange), medium (green), and low (blue) expres-sion levels. Dendrograms represent the clustering of the expression data obtained at 37 °C. Lateral bars represent which expression cluster each variant belongs to at the indicated temperature for comparison.

5.77 Wm$^{-2}$ light induction respectively; see Supplementary Table 13) as well as the negative control (from 0.3096 to 0.329 Hill slope in the dark and with 5.77 Wm$^{-2}$ light induction respectively; see Supplementary Table 13). A lower hill slope corresponds to a slower growth rate (compare growth curves at 30 °C in Supplementary Fig. 21, 37 °C in Supplementary Fig. 22 and 40 °C in Supplementary Fig. 23). As a result, given similar starting ODs, also the time for the culture density to reach half maximum ($T50$) was shifted to later time points correspondingly and causing an up to 2 h delay for the two highest expressing variants pMag P42T T69M and pMag R52L, while for both the negative control and the wild type, no delay in reaching the $T50$ point was detected when comparing dark to 5.77 Wm$^{-2}$ light induction conditions (see Supplementary Table 13 and Supplementary Table 14). These growth differences reflect that these four variants show the highest expression levels of all variants and that expression at 30 °C was highest in the three incubation conditions (30 °C, 37 °C, and 40 °C). This behavior can be explained by the strong mCherry reporter overexpression, which led to metabolic burden effects that result in reduced growth in light-induced conditions.

Alongside manual comparison and the extraction of relevant enzyme parameters as mentioned above, we also performed a principal component (PC) analysis of the datasets. The first PC could explain over 95% of the variance in the flow cytometry data (Supplementary Fig. 24A–C). In the case of the spectrophotometry data acquired over time, the first PC could explain only 62% of the variance. We reasoned that if the data were not scaled before the PC analysis, this would prevent giving excessive weight to the fluorometric data obtained at low OD values, when the bacterial cultures are starting to grow and gene expression could not be detected yet. Indeed, using non-scaled data, the first PC could explain over 95% of the variance in the data (Supplementary Fig. 24D–F). In both sets of data, all the variants were nicely distributed along the PC1 (x-axis) according to their expression level (compare with Fig. 2A, B and Fig. 3A, D). Visual inspection of the PCA plots suggested that the variants could be clustered by expression level. We performed hierarchical clustering (HC) of the complete flow cytometry dataset of the different light intensities and the complete dataset of the spectrometry data over time and at the different light intensities using the Ward clustering method with Euclidean distances[50] (Fig. 3E, F). Four clusters were identified that contained variants with very high (red), high (orange), medium (green), and low (blue) expression profiles (Fig. 3E, F). The negative control (NC) not harboring the Magnets constructs was in the low expression cluster, accordingly (reporter expression time courses, shown in Supplementary Fig. 25). Visual inspection of the PC and HC cluster analyses showed that they were in agreement. Therefore, this clustering was used to analyze data gathered at different incubation temperatures during the characterization of the mutations. Despite the two datasets using different measurement methods (single-cell by flow cytometry and population by fluorometry) and time considerations (single timepoint and expression measured over time), only slight changes were observed when comparing the corresponding dendrograms at the screening temperature (37 °C) (Supplementary Fig. 26). The red and orange clusters contained variants with mutations in the pMag, apart from M48V N100L, while the green and blue clusters contained variants with mutations both in pMag and nMag.

### nMagHigh1 variants with increased light sensitivity and improved light activation at 37 °C

The nMagHigh1 variants L39P and V160M were located in the medium expression (green) cluster (Fig. 3 E) and showed an increased light sensitivity, described with a decrease in the half-maximal light intensity ($I50 = 80$ and 83%, respectively, compared to WT, Fig. 2A, Supplementary Table 2). We also observed a higher light activation, presumably due to improved dimerization properties in these variants, visible through the increased expression at a saturating light intensity

($t = 151$ and 148% compared to WT respectively, Fig. 2A), while also showing increased basal expression ($b = 150$ and 148% compared to WT, Fig. 2 A). Interestingly, a double mutant also containing a mutation at position 39 (L39R) with a second mutation T38A located in the low expression cluster (blue) showed an even further increased light sensitivity compared to the single mutant ($I50 = 68$% compared to the WT). Together with the lower light activation, visible through reduced basal- and light-induced reporter expression ($b = 84$%, $t = 77$%, Fig. 2A), this indicates that expression levels and light sensitivity can be tuned separately. In addition, we identified the double mutant M48V N100L, which shows a dramatically increased light sensitivity ($I50 = 50$% compared to the WT, light dose-response curves, and expression at different intensities over time of WT and M48V N100L, Fig. 2C, D). In addition, both basal- and light-induced expression are increased in this variant ($b = 296$%, $t = 267$% compared to the WT, Fig. 2A). Accordingly, this variant clustered in the high expression group (orange).

### nMagHigh1 variants with decreased light sensitivity at 37 °C

nMagHigh1 variants with decreased light sensitivity clustered in the low-expression group (blue). This includes mutations A143S and V147M (with an $I50 = 143$ and 162%, respectively, compared to the WT, Fig. 2E, Supplementary Table 2) that displayed expression levels comparable or slightly reduced compared to the WT ($b = 91$ and 89%, $t = 95$, and 95% respectively, Fig. 2C). Also here, expression level and light sensitivity were tuned separately. In addition, double mutant I135V K137R showed an increase in the half-maximal light intensity ($I50 = 158$% compared to the WT), a reduced dark expression ($b = 88$%), and a slight increase in the maximal light-induced expression ($t = 111$%, n.s., Fig. 2A).

### nMagHigh1 variant with shifted light activation at 37 °C

Variant K137E showed a light sensitivity that is comparable to the WT ($I50 = 98$%, Fig. 2F). However, the reporter gene expression was upshifted for both the dark- as well as the light-induced expression ($t = 128$%, $b = 133$% compared to the WT, Fig. 2C, Supplementary Table 2). Interestingly, this variant also clustered in the low expression group.

### pMag variants with increased light sensitivity and light activation at 37 °C

All pMag variants, except for positions N100, were unique (Fig. 2B) compared to the ones previously described for nMagHigh1 (Fig. 2A). While N100S was found in pMag, a mutation at position N100 was also found in nMagHigh1 double mutant M48V N100L. pMag variants N100L, N100K, and N100S as single mutations were all identified during site saturation mutagenesis (see below). All variants R52L, T69M, T83M, N100K, N100L, N100S, K119N, M165T, S178Y, P42T T69M, and P66S Y94N showed an increased light sensitivity of up to 47% compared to the WT ($I50 = 79, 50, 47, 71, 59, 89, 78, 50, 64, 52$ and 83% respectively, Supplementary Table 4). The increase in light sensitivity was most pronounced in variants T69M, T83M, N100L, M165T, and P42T T69M, all belonging to the high or very high expression clusters ($I50 = 50, 47, 59, 50$, and 52% respectively, Fig. 2G–I, Supplementary Table 4). In addition, for all variants except for K119N (located in the low expression cluster), the expression levels were increased, which was most pronounced (higher than 300%) in variants T69M, N100L, and P42T T69M up to 540% ($t = 326, 540$ and 517% respectively) and comparably less increased for the basal expression ($b = 257, 362$ and 422% respectively). In contrast, variant K119N showed a light activation and thus expression levels comparable to the WT enzyme ($b = 121$%, $t = 105$%) while having an increased light sensitivity ($I50 = 78$%, Fig. 2J), again indicating that these properties can be tuned separately. Comparing both T69M variants, the single (high expression cluster, orange) and double mutant with P42T (very high expression cluster, red), light sensitivities are similar ($I50 = 50$ and 52%), while

expression levels are upshifted in the double mutant ($b = 257$ and 422%, $t = 326$ and 527% respectively), indicating an even further improved dimer assembly through the additional P42T substitution.

## Dynamic off-switching at 37 °C

The observed changes in light sensitivity and expression level could potentially also be caused by changes in the dynamic off-kinetics, although this was not specifically screened for. To investigate these effects, a pseudo-time course was performed (Supplementary Fig. 27). Cells were induced with constant saturating light intensity (3.85 W m$^{-2}$) for 120 min before turning off the light. nMagHigh1 variants with increased (M48V N100L, $ISO = 50$%), similar (K137E, $ISO = 98$%) and decreased (I135V K137R, $ISO = 158$%) light sensitivity at 37 °C were chosen (Fig. 2D, F, Supplementary Table 2-Supplementary Table 3). In addition to their difference in light sensitivity, M48V N100L also shows the overall highest expression of all identified nMagHigh1 variants. For pMag, N100L was chosen along with the two variants containing either T69M, either alone or in combination with P42T. N100L and P42T T69M are the two highest expressing pMag variants at 37 °C (Fig. 2H, I, Supplementary Table 4-Supplementary Table 5). All variants, except for T69M, follow the WT dynamics closely, suggesting no major change in the off-kinetics. The variant with mutation T69M shows sustained activity after the light input is removed, which might be due to slower dark-state reversal. A mutation at the same position (T69W), which is located at the dimer interface, was previously described to cause a constitutively active homodimer[41]. One hypothesis is that T69M renders the magnet domain either constitutively active or slower-reverting, making inactivation dependent solely or mainly on the complementary Magnet domain, a hypothesis that could be mechanistically investigated in follow-up studies.

## Site saturation mutagenesis at hotspot positions T69 and N100

Random mutagenesis can be used to identify important amino acid residues and positions. We chose the two sites T69 and N100 in the very high expression clusters (red) for further investigation through site saturation mutagenesis (SSM) as they appeared multiple times during our screening both as single and as double mutants and in addition showed highly interesting properties. T69M, as a single mutant, as well as a double mutant with P42T, was one of the variants with the highest low-light sensitivity in the pMag photosensor. N100 was the only residue in which variants were selected in both the nMagHigh1 as well as the pMag libraries, and in both cases increased the light sensitivity of the photosensor. As both mutations were found in the pMag photosensor, we also used this domain to construct SSM libraries either using the "22-codon trick"[51] or as a "small intelligent"[52] library. From these libraries, 93 randomly selected colonies were picked per 96-well plate, as well as three WT controls in the same plate and incubated either in the dark or induced with low-intensity light as previously described.

SSM of position T69 was performed with a reduced "small intelligent"[52] library with NDT, VMA and TGG for position T69, but lacking the ATG, as T69M was already identified. Due to the large number of variants to screen, we only chose two light intensities (subsaturating 0.95 W m$^{-2}$ and saturating 3.78 W m$^{-2}$ for WT Opto-T7RNAP) and dark controls for the screening. Although this does not allow for the fitting of the dose-response model and identification of parameters $b$, $t$, and $ISO$, it still enables one to identify variants that show higher light sensitivity or activation than what was previously identified. As shown in the inset in Supplementary Fig. 28A, T69, shown as red spheres, lies in the interface of the photoactivated VVD dimer. As a result of the SSM, only T69W was identified to perform significantly better at low light induction compared to the WT, while T69L and T69F performed similarly to the WT. However, variant T69M found during the screening outperformed T69W for mCherry expression in low as well as high light induction conditions

(Supplementary Fig. 28B). Interestingly, despite T69W was reported to form VVD dimers in the dark[41], here we found no difference in mCherry expression between the WT and the T69W in the dark, noting that the mutation is only present in one of the heterodimerization domains. T69M showed a 2.4-fold higher light activation when comparing low light expression levels to the WT expression level (Supplementary Fig. 28C). However, also the activation of T69W was improved 1.7-fold. For both T69M and T69W, but also for the leucine, phenylalanine and tryptophan substitution, the fold change at low light induction was improved by 2.8-, 1.5-, 1.6- and 1.3-fold, respectively (Supplementary Fig. 28D). In general, all mutations improving mCherry expression are amino acids with hydrophobic side chains and therefore might play a role in the dimer formation, while substitutions that decreased the function of the photoregulator were mostly hydrophilic or polar e.g., aspartic acid, arginine, or serine (Supplementary Fig. 28A–D). In summary, no variant outperforming T69M, obtained during the initial screening, could be identified by site saturation mutagenesis of this residue.

SSM of position N100 was performed using the "22-codon trick"[51]. The position is indicated by red spheres, located at the surface of the protein close to the Ncap (Supplementary Fig. 28E inset). During the screening, we already identified N100L in combination with M48V and N100S. N100L showed the highest expression level at low light induction (Supplementary Fig. 28E, F), which was higher than the levels obtained with the double mutant M48V N100L (Fig. 2A, B). The light activity of the single mutant was increased 9.9-fold for the expression at the low light intensity (Supplementary Fig. 28G), compared to 5.7-fold for the double mutation, suggesting that M48V might have a slightly detrimental effect which is overcompensated by N100L in the double mutant compared to the WT. N100L also showed an increase in the fold change at low light intensity of 3.6-fold. Further, methionine, lysine, and arginine, along with the previously identified serine, showed increased light activation of 3.2-, 2.9-, 2.9-, and 2.5-fold, respectively. The dark-to-light fold change was also improved in all of these variants at low light intensities with the highest improvement of 3.6-fold for N100L and slightly improved or similar to the WT at high light intensities (Supplementary Fig. 28H). The different properties of improving mutations, ranging from polar via positively charged to hydrophobic side chains, do not allow for clear mechanistic conclusions and require additional assessment in future studies. Also, for position N100, no further variant with a higher light sensitivity and activation than N100L could be identified by site saturation mutagenesis of this residue, revealing the benefits of a random mutagenesis coupled to a high-throughput screening compared to a rational design strategy.

## Light-induced gene expression at 30 °C, 37 °C, and 40 °C

We performed the same functional characterization as described before (reporter expression at different light intensities) also at lower (30 °C) and higher (40 °C) temperatures than the standard 37 °C cultivation conditions. This should aid in further investigating the performance of the Opto-T7RNAP and the discovered variants in different culture conditions, as well as its effect on the mutations themselves. As described for the experiments at 37 °C, characterization was done by flow cytometry and over time was measured through fluorescence spectrophotometry. In general, we observed an approximately 4.5-times higher maximal mCherry fluorescence ($t = 128{,}601$ arb. units, Supplementary Table 6) in cells containing the original Opto-T7RNAP*(563) regulator at 30 °C compared to 37 °C ($t = 28{,}360$ arb. units, Supplementary Table 2), which might be caused by a decrease in protein stability and/or in dimerization ability at the higher temperatures. In addition, the light intensity required for half-maximal activation decreased 3.9-fold (from $ISO = 0.67$ W m$^{-2}$ at 37 °C to $ISO = 0.17$ W m$^{-2}$ at 30 °C). Therefore, experiments were performed at a lower light intensity range for experiments performed at 30 °C (from 0

to 0.96 W m$^{-2}$) compared to the experiments at 37 °C (from 0 to 5.77 W m$^{-2}$). Accordingly, similar effects were observed for experiments performed at 40 °C. At this temperature, the highest used light intensities (5.77 W m$^{-2}$) did not lead to a saturating reporter expression (Supplementary Fig. 9, Supplementary Fig. 19). To avoid additional heating of the samples caused by the light induction, we did not further increase the light intensities. As increased light intensities would be necessary to reach saturation, which would be required to mathematically fit the dose-response curve, we directly compare the fluorescent signal of the reporter expression in the dark and in the light intensity that led to saturation at 37 °C (3.85 W m$^{-2}$). In comparison to the reporter expression at 37 C, the higher temperature (40 °C) led to a 114-fold reduction in reporter expression with 3.85 W m$^{-2}$ light induction (28,854 arb. units and 252 arb. units respectively, compare Supplementary Table 2 and Supplementary Table 10), while dark expression was reduced around 6.4-fold (517 arb. units and 82 arb. units respectively). To summarize, this pre-characterization of the wild-type Opto-T7RNAP system showed that light-induced reporter expression and light sensitivity were highest at 30 °C and decreased as the temperature was increased to 37 °C and 40 °C. As described above at 37 °C, analogous PC analysis showed that all the variants were nicely distributed along the PC1 (x-axis) according to their expression level also at 30 °C and 40 °C (Supplementary Fig. 24A, C, D, F). Analysis of the number of clusters using the silhouette algorithm[53] and the data at all temperatures suggested an optimal number of clusters of 4-5 or 4, depending if the dataset was single-cell or population. Four clusters were selected and visual inspection of the HC analyses suggested that both types of data were in agreement.

### Variant characterization at 30 °C
As observed for the WT, all variants showed an increased reporter expression at the lower expression temperature (Fig. 3A, B) compared to 37 °C. Interestingly, due to the overall higher expression at 30 °C (Fig. 3A, B), the NC was the only variant in the low expression cluster, while at 37 °C and 40 °C the low expression cluster contained 6-7 and 8-11 variants, respectively. From the single-cell expression data, all the members of the medium expression cluster, except V160M, L39P, N100S, and N100K moved to the low expression cluster at 37 °C and 40 °C. In most cases, the general trends observed for the different variants (Supplementary Table 6) are comparable to 37 °C (Supplementary Table 2). Interestingly, nMagHigh1 variant A143S, which showed a reduced light sensitivity at 37 °C ($I50 = 143\%$ compared to the WT, Supplementary Table 2), displayed a similar sensitivity as the WT at 30 °C ($I50 = 98\%$, Supplementary Table 6). Additionally, pMag variant R52L showed a dramatically increased expression level at 30 °C, which was similar to pMag double mutant P42T T69M ($t = 206$ and 207% respectively compared to the WT, Supplementary Table 8) in addition to the increased light sensitivity ($I50 = 63$ and 56% respectively, Supplementary Table 8). In comparison, at 37 °C, the expression of R52L was still increased compared to the WT ($t = 275\%$, Supplementary Table 4) but significantly lower than P42T T69M ($t = 517\%$) or N100L ($t = 540\%$). Accordingly, variant R52L moved from the very high expression (red) cluster at 30 °C to the high (orange) at 37 °C and medium expression (green) cluster at 40 °C in both datasets, while N100L moved from the high at 30 °C to the very high expression cluster at 37 °C (Fig. 3E, F).

### Asymmetric changes for dark and light-induced expression
nMagHigh1 variants V147M and I135V K137R both led to a more pronounced decrease in the basal expression at 30 °C ($b = 47$ and 41% respectively, Supplementary Table 6) compared to the expression at 37 °C ($b = 89$ and 88%, Supplementary Table 2), whereas the relative maximal expression was similar in both conditions ($t = 97$ and 106% at 30 °C and $t = 95$ and 111% at 37 °C). The contrary was observed for nMagHigh1 variant M48V N100L, where light-induced expression was

similar and the basal expression increased compared to the WT at 30 °C ($t = 109\%$, $b = 271\%$, Supplementary Table 6), but increased for both conditions at 37 °C ($t = 267\%$, $b = 296\%$, Supplementary Table 2) whereas the dramatic decrease in light sensitivity was comparable for both temperature conditions ($I50 = 50\%$ at 37 °C; $I50 = 52\%$ at 30 °C). For this variant, the extremely low light intensity of 0.09 W m$^{-2}$ led to half-maximal activation of gene expression. Similarly, pMag variants T69M, T83M, N100K, N100L, and P42T T69M showed a lower increase in the maximal expression at 30 °C compared to 37 °C ($t = 132, 125, 88, 165,$ and 207% at 30 °C; $t = 326, 270, 190, 540,$ and 517% at 37 °C, respectively), while the change in the basal expression at both temperatures was similar or comparably less ($b = 260, 245, 184, 253,$ and 468% at 30 °C; $b = 257, 232, 145, 362,$ and 422% at 37 °C). For all pMag variants, an increased light sensitivity was observed at 30 as well as 37 °C (Supplementary Table 8, Supplementary Table 4).

### Variant characterization at 40 °C
Similar to the WT, the expression of all variants was dramatically reduced at 40 °C compared to 37 °C (Fig. 3C, D, Supplementary Table 10-Supplementary Table 12). However, all variants with increased light sensitivity and increased light activation (reporter expression) at 37 °C, also showed an increased reporter expression compared to the WT in all light conditions, which might suggest an improved thermostability. The largest increase in expression compared to the WT was observed with nMagHigh1 variant M48V N100L, and pMag variants T69M, N100L, P42T T69M which showed an increase in reporter expression of 412, 407, 782, and 781% respectively at the highest light intensity (Supplementary Table 10-Supplementary Table 12), but only variant N100L moved from the high at 37 °C to the very high expression group at 40 °C to cluster together with P42T T69M. Variant nMagHigh1 T38A L39R, which displayed increased light sensitivity at both 30 °C and 37 °C ($I50 = 75\%$ at 30 °C; $I50 = 68\%$ at 37 °C), showed decreased expression levels at all temperatures (compare Supplementary Table 2 and Supplementary Table 10).

### Population data over time supports single-cell data measured at a single time point
The results obtained with the clustered spectrophotometry data over time were highly similar to the single-cell data as described above, and confirmed that the behavior of the variants is not restricted to one single time point at low cell density, but rather consistent up to the early stationary phase of growth. To further analyze the expression at higher cell density, we fitted the fluorescent data to light dose-response curves at one specific time point at which the variants reached the highest fluorescence levels. While in the single-cell analysis, cells were in the exponential growth phase at low cell density, the time point with the highest fluorescence in the population data was reached at the end of the exponential phase - beginning of the stationary phase. Despite this difference in growth stages, results were comparable. For example, nMagHigh1 variants L39P, T38A L39R, and V160M showed increased light sensitivity through a decrease in the half-maximal light intensity at 37 °C ($I50 = 70, 70, 52\%$ respectively, compared to the WT; Supplementary Table 3). Also in these experiments, nMagHigh1 K137E showed a light sensitivity that is comparable to the WT ($I50 = 102\%$). As observed in the single cell data, also here all pMag variants revealed an increased light sensitivity ($I50 = 79, 60, 55, 88, 71, 88, 78, 63, 58, 55$ and 83% compared to the WT for R52L, T69M, T83M, N100K, N100L, N100S, K119N, M165T, S178Y, P42T T69M and P66S Y94N respectively; Supplementary Table 5). The previously described reduced light sensitivity at 37 °C and similar sensitivity as the WT at 30 °C for nMagHigh1 variant A143S was also observed in the population dataset ($I50 = 160\%$ at 37 °C, $I50 = 117\%$ at 30 °C; Supplementary Table 3, Supplementary Table 7). Also, the asymmetric changes for dark and light-induced expression of nMagHigh1 variants V147M and I135V K137R, which led to a more pronounced decrease in

the basal expression at 30 °C compared to the expression at 37°C and more similar maximal expression in both conditions (see discussion above) could be observed in the population dataset (V147M: $b$ = 103%, $t$ = 86%, $ISO$ = 167% at 37 °C and $b$ = 47%, $t$ = 99%, $ISO$ = 143% at 30 °C; I135V K137R: $b$ = 111%, $t$ = 82%, $ISO$ = 178% at 37 °C and $b$ = 51%, $t$ = 98%, $ISO$ = 124% at 30 °C; Supplementary Table 3, Supplementary Table 7).

Altogether, the hierarchical clustering, as well as the comparison of parameters, show that the mutations led to similar behaviors both for expression measured at a discrete time point during the early logarithmic growth phase, as well as for gene expression over time. This is an important characteristic for future applications of this optogenetic system.

## Discussion

### Features of identified mutations

In this work, we have identified a set of 14 single and five double amino acid substitutions in the light-inducible Magnets domains after a random mutagenesis screen. All mutations show changes in their light sensitivity and/or their expression levels. Interestingly, we found that properties, such as light sensitivity and light activation could be changed independently. For example, pMag N100L shows both increased reporter expression and light sensitivity compared to the WT, while for nMagHigh1 T38A L39R, which was also more light-sensitive, we observed a decreased reporter expression. Variant K137E in nMagHigh1, in turn, showed an upshifted dark- as well as light-induced expression, while maintaining a similar light sensitivity compared to the WT. Increased light sensitivity variants thus require lower light intensity for full induction, which is beneficial when light toxicity or light penetration into a sample is problematic (e.g., in large culture volumes). The increased overall expression level of some mutants is especially interesting for systems in which the setpoint for dark- and light-induced expression cannot easily be altered. Through the use of these mutations, the setpoints can be genetically fixed at different levels.

While the goal of this study was to demonstrate an approach for easy photoreceptor tunability, it would be of interest for further studies to test combinations of mutations, especially if certain properties of different variants are desired. Although mutations might not behave additively, we found several identical amino acid substitutions both in single- as well as in double-mutant variants. An example is double mutant P42T T69M, which shows a similar $ISO$ to the single mutation T69M, but an increased expression level, suggesting that such a combinatorial screening for additive effects of these mutations might further enhance certain properties. In addition to combining mutations within the individual domains, combinations of mutations in the different heterodimerization domains should also be investigated in future studies. This could allow for further tuning of the individual properties.

To better understand how it is possible that light activation level or light sensitivity could be so easily changed, the context of the native protein origin must be taken into consideration. The VVD photodimerization system originates from *Neurospora crassa*, where it is involved in photoadaptation and the circadian clock. It was previously reported that the VVD protein is temperature-regulated, degrading faster at higher temperatures to ensure a stable clock in a wide range of physiological temperatures[54]. Since the Opto-T7RNAP expression system is mainly used at 37 °C in both bacteria and mammalian cell cultures, we performed our screening at this temperature. Thus, improved thermostability might be one feature that was optimized for along with improved dimer assembly or light-mediated allosteric changes within the protein. Regarding thermostability, different phenomena could be observed. For example, pMag variant R52L showed the most pronounced expression level increase at 30 °C, where it was the highest expressing variant together with double mutant P42T T69M (Fig. 3B). At 37 °C its expression level was still 275% higher than

the WT, however, it was outperformed by several other variants (T69M, N100L, P42T T69M with $t$ = 326, 540, and 517% respectively, Supplementary Table 4). The same variants also showed a higher expression than pMag R52L at 40 °C (Fig. 3D, F). A similar trend was seen with variant P66S Y94N, which belongs to the high expression cluster at 30 °C, and the improvement of gene expression in comparison to the WT decreased at 37 °C and was comparable to the WT at 40 °C (Fig. 3E, F), indicating that thermostability might not be the major contribution of the mutation for the changes in properties. The opposite was observed for pMag N100L, which was outperformed by R52L and P42T T69M and was on par with P66S Y94N at 30 °C. At 37 °C and 40 °C however, N100L was the highest expression variant, indicating an improvement in temperature stability. In contrast to these increasing or decreasing expression levels with temperature, pMag variant P42T T69M was consistently one of the highest expressing variants at all temperatures, which might indicate improved temperature stability as well as dimerization properties. Comparison with the findings of Vaidya et al. [41] shows that a mutation in the same residue, in this case T69, could have different effects depending on whether the VVD photoregulator is used for homodimerization or through Magnets mutations as a heterodimerization system. Although it leads to constitutive dimerization in the dark and light in the homodimerization system[41], it led to improved light sensitivity and dimer assembly in our study.

### Structural comparison of mutations

Our goal was to develop a strategy that enables quick and easy parameter tuning of the photodimerization domain. By not restricting the engineering to specific functional sites of the protein, such as the FAD-binding pocket or the dimer interface, we obtained mutations in different compartments of the domain. In general, the VVD photoregulator comprises a LOV Per-Arnt-Sim domain with an N-terminal cap region (Ncap; residues 37–70) and a loop that accommodates the flavin cofactor[27]. Light induces a covalent cysteine-flavin adduct between the C108 cysteine thiol and the flavin C4a position, and induces conformational changes that propagate to the N-terminal helix and the protein surface, leading to the release of the N-terminal part from the protein core and forming a symmetric dimer via hydrophobic amino acids. Return to the dark state is caused by the scission of this thioether bond. The flavin ring is bound in a pocket formed by two helices (Eα and Fα) and three strands of the central beta-sheet (Aβ, Hβ, and Iβ) at the end of a water channel, and the structure contains an 11-residue FAD loop at the surface of the protein[27].

The single mutations found are located in all functional parts of the two photosensory domains (shown as red spheres in Supplementary Fig. 29A). For visualization purposes, we used the solved structure of the light-induced VVD dimer (PDB code: 3RH8). From the Ncap mutations, T38A, L39P, P42T are located in the N-terminal "latch" (amino acid residues 37 - 44) that wraps around the domain (Supplementary Fig. 29C). Mutations M48V and R52L are located in the interface of the dimer within the subsequent Aα helix, and P66S and T69M are located in the hinge region of the dimer interface. Several single mutations (T38A, L39P, R52L, T69M, and T83M) all show hydrophobic properties, which might aid in dimer formation. Interestingly, T69W was previously described as facilitating intersubunit contact[41]. In this study, T69W caused VVD homodimerization in both the dark and the light state. In our work using the Magnets, with both variants T69M and T69W, we still observed light inducibility, resulting in even higher mCherry expression levels than the WT. M165T, S178Y, T83M, and M179V are located at the flavin harboring water channel (Supplementary Fig. 29D, E). Specifically, M165T, S178Y and T83M are in proximity to the flavin isoalloxazine ring (<3.8 Å for the original residues in VVD), which might aid in positioning of the chromophore for the formation of the cysteine-flavin adduct and subsequent

structural rearrangements. T83M at the entrance of the water channel harboring the flavin chromophore and K119N in the FAD loop (Supplementary Fig. 29E) were both characterized as single mutations and might alter chromophore binding and positioning, again allowing for cysteine-flavin adduct formation at lower light intensities. To gain more definite answers, x-ray crystallography, and UV-vis spectroscopy[55] could shed more light on the underlying structural and mechanistic functions of these amino acid exchanges. This is out of the scope of our engineering-driven approach and should be investigated in future studies.

Zhou et al identified five features that better define the differences between the active, inactive, and transition states. Of those five features, the most important ones were the distances T38-G105 and T38-K119[56]. Interestingly, we found mutations in two of those residues, both reducing the length of the side chain (T-to-A in 38 and K-to-N in 119), for which we observed reduced light sensitivity.

## Comparison to other VVD mutations

Our findings also revealed interesting connections and additions to previously known mutations. An intriguing example was identified with variant pMag R52L. Previously, VVD variant I52C was reported as having increased homodimer-forming efficiency in both the dark and light state[20]. This position was then used to transform the homodimeric VVD protein into the heterodimeric nMag/pMag Magnets system, in which I52R together with M55R were used for the "positively charged" pMag[26]. Our mutagenesis revealed that, in combination with nMagHigh1, pMag variants in which the positively charged arginine is changed to leucine, and thus back to an amino acid with a similarly hydrophobic sidechain as with the initial isoleucine, increase both expression and light sensitivity. In our characterization, this change increased both the light-induced and basal expression as well as the light sensitivity at all tested conditions (Supplementary Table 4, Supplementary Table 5, Supplementary Table 8, Supplementary Table 9, Supplementary Table 11).

Residues M135 and M165 are in contact with the flavin ring. Substitution of those residues with isoleucine resulted in variants that remained in the 'on' state tenfold longer[42], slowing the photocycle[57] of *N. crassa*. During the Magnets development, these mutations M135I M165I were identified to further increase dimerization efficiency. The Opto-T7RNAP version used for screening contained these mutations in the nMag (nMagHigh1); however, we did not include them in the pMag domain, as the combination of nMagHigh1 and pMagHigh1 shows increased dimerization in the dark[26]. During screening for increased expression levels, we identified a variant that contained both mutations, and thus it was identical to pMagHigh1.

Another interesting finding was T69M, which appeared both as a single mutation and a double mutation together with P42T in pMag. Vaidya et al. found that in VVD, T69W forms dimers in the presence and absence of light. They attributed this to facilitated inter-subunit contacts that overcame light-promoted conformational switching, as previously described. In the Magnets, T69L was found to improve hydrophobic interactions[45]. T69L, together with S99N, M179I, were identified from comparison with VVD domains from thermophilic fungi[45]. It was hypothesized that T69L improves hydrophobic interactions, M179I in the hydrophobic core improves packing, and surface-exposed S99N optimizes hydrogen bonding and secondary-structure preference. Our site saturation experiments on position T69 identified T69M as the variant that showed the highest expression levels both with high and low light induction, although the expression of T69W was also increased in both cases (Supplementary Fig. 28B). Thus, while T69W might lead to light-independent binding in the homodimeric VVD protein, our findings suggest that the heterodimeric interactions in the Magnets allow for the change of one component, which increases both the expression level as well as light sensitivity.

Our characterization further revealed that multiple different side-chain exchanges of N100 led to both increased expression levels and light sensitivities, with N100L showing the highest changes. N100R was previously reported to allow for stronger dimerization at 37 °C due to improved helical preference, which was also rationally identified due to comparison with VVD variants from thermophiles[45]. In our study, N100L led to the highest expression at both 37 °C and 40 °C through improved thermostability. Our site saturation mutagenesis confirmed that the previously identified N100R variant also shows increased expression compared to the wild type, however, it was outperformed by N100L (Supplementary Fig. 28).

## Beyond bacterial applications - variants enhance gene expression in mammalian cells

Variants with increased light sensitivity and boosted gene expression might be useful to address two major challenges of optogenetics using the mOptoT7[58] in mammalian cells. First, optogenetic systems with increased light sensitivity require lower-intensity light for induction, thus reducing potential phototoxic effects[46], which is especially important during long-term experiments. Second, although the mOptoT7 has the unique advantage of being an orthogonal transcription system, the overall expression level of the mOptoT7 is lower than other optogenetic systems in mammalian cells, as shown by the direct comparison with GAVPO[58]. Thus, elevating the light-induced expression level widens the applicability of the mOptoT7.

For this, we performed an explorative study with a reduced set of mutations and incorporated them into version 1 of the mOptoT7. We used these mOptoT7 versions for light-induced expression of mRuby3 from a separate reporter plasmid (Supplementary Fig. 30A). An IRES2 sequence and a polyA tail were used for enhanced expression in mammalian cells[58]. A constitutively expressed mCitrine plasmid served as a transfection control. HEK293T cells were transfected as described in the Methods section and measured through flow cytometry after 24 h of constant light induction at a previously characterized light induction condition (0.8 and 0.1 W m$^{-2}$)[58] or incubation in the dark. We observed a modestly increased expression level at the lower light intensity with nMagHigh1 variants L39P, N100L, and N100L M48V compared to the WT. In particular, the M48V N100L variant showed an approximately 3-fold upshift of normalized reporter expression compared to the wild type at all dark and light conditions (Supplementary Fig. 30B). These experiments indicate that the mutations characterized in *E. coli* might be especially interesting if low-intensity light needs to be applied due to phototoxicity/light-application limitations or if a generally high expression of a gene of interest is required. These additional experiments highlight the potential to investigate these variants in other organisms for specific applications.

## A versatile approach for tailoring optogenetic tools to diverse applications

Overall, our results show that the Opto-T7RNAP transcription system creates an excellent phenotype-genotype linkage that can be used to adapt various properties of this and possibly other heterodimerization systems toward their actual application in research or biotechnological processes. We have previously identified mutations/residue positions and have now identified new ones, along with hotspots. We challenged this approach by screening for variants with improvements in the most fundamental feature of photoregulators, their light sensitivity and activation. After just one round of mutagenesis, variants with mutations within all the important structural compartments of the photoregulator, the Ncap, the PAS core, as well as the FAD-loop were identified. The mutations that lead to increased light sensitivity and activation were distributed all over the photosensory domain, highlighting the benefit of random mutagenesis of the full photosensor compared to rational selection of residues at important sites. We showed that selected mutations can also be used in mammalian cells to

Article

| | | | | | | | |
|---|---|---|---|---|---|---|---|
| Light sensitivity (I50) | ↑ | ↑ | ↑ | ↑ | ↑ | ↓ | ↓ |
| Maximal expression (t) | = | ↑ | ↑ | ↑ | ↑ | = | = |
| Fold induction (t/b) | = | = | ↓ | ↑ | ↑ | = | ↑ |
| Thermostability | = | ↑ | ↓ | ↓ | ↑ | = | = |
| Variants | T38A L39R K199N | L39P V160M M48V N100L P66S Y94N | M165T | R52L | N100L P42T T69M T69M T83M N100K S178Y N100S | A143S V147M | I135V K137R |

Fig. 4 | Summary of variant properties for application purposes. Variants are classified according to which properties are equal (=), enhanced (↑) or weakened (↓) at 37 °C.

overcome some of the limitations of blue-light-based optogenetic tools: light toxicity and limited gene expression levels. This approach can also be used to screen for other properties, such as lower dark-state assembly, improved light-induced dimerization at saturating light induction, increased dark-to-light fold change, or even dynamic features, such as increased or decreased dark-state dissociation rates by varying the light input before sorting. A crucial requirement for successful screening campaigns, especially for features such as induction at sub-saturating light intensities, is the relatively high fold-change of the Opto-T7RNAP. Thus, mCherry expression levels of cells containing the original Opto-T7RNAP*(563) were still sufficiently different from dark controls to screen with sub-saturating light induction, which allowed us to distinguish desired variants from mutations that inactivated the function of the regulator. This also eliminated the need for multiple enrichment steps across the different induction conditions, resulting in a broad set of variants. While we screened for changes in the light sensitivity and expression levels of the photo-dimerization domains, the same approach can be undertaken to screen for other properties. This is essential for perfectly adjusting photosensors to specific applications. In addition, the variants we identified for the Magnets domains, which are based on the VVD photoregulator, will find immediate use in the many existing designs that utilize this photosensor and will significantly expand the applicability of these domains and the optogenetic toolbox (See Fig. 4 for a summary of the most promising variants classified according to their properties for specific applications).

## Methods

### Bacterial strains and media
E. coli Top10 was used for all cloning. For characterization, we used E. coli strain AB360[8]. The strain contains the transcription factor AraC, whereas arabinose-metabolizing genes araBAD are deleted, and the mutated permease lacYA177C for titratable arabinose regulation. Plasmids were transformed using a one-step preparation protocol of competent E. coli for the transformation of plasmids in testing strains[59] or made electrocompetent for subsequent electroporation[60]. E. coli CloneCatcher™ DH5G Gold Electrocompetent was used for high-efficiency transformation of mutagenesis libraries following the manufacturer-provided protocol (Genlantis, Inc.).

An autoclaved LB-Miller medium was used for strain propagation. Sterile-filtered M9 medium (M9 Minimal Salts 5X, Sigma Aldrich) supplemented with 0.2% casamino acids, 0.4% glucose, 0.001% thiamine, 0.00006% ferric citrate, 0.1 mM calcium chloride, 1 mM magnesium sulfate was used for all gene expression experiments. Antibiotics (Sigma-Aldrich Chemie GmbH) were used as necessary for plasmid maintenance at concentrations of 100 µg/mL ampicillin, 17 µg/mL chloramphenicol, and 50 µg/mL kanamycin.

### Plasmids and genetic parts
Plasmid pAB150 containing the Opto-T7RNAP*(563) as well as the reporter plasmid pAB50 with mCherry under T7 promoter expression control are taken from our previous study[8]. All primers used were purchased from Microsynth AG, Switzerland and are listed in Supplementary Table 15. The plasmid with lower TIR was constructed by using the RBS Calculator V.2.0[48] and inserted via PCR amplification of mCherry with primers oAB819 and oAB507 from template pAB50, digested with BamHI and KpnI and ligated into the BamHI and KpnI digested backbone. Primer and plasmid sequences are described in Supplementary Table 15.

Error-prone PCR of Magnets domains was performed through amplification of nMagHigh1 and pMag from pAB150 with primers pairs oAB734/oAB736 and oAB810/oAB744 respectively and error-prone Pfu DNA Polymerase containing D141A, E143A mutations[61], a gift from Dr. Luzius Pestallozi (DBSSE, ETH Zurich). We used template concentrations of 2.5, 5, and 10 ng to reach variable mutation rates. As backbone we used pAB150 amplified with Phusion HF polymerase (Phusion flash HF PCR master mix, Thermo Scientific) and primer pairs oAB589/oAB446 and oAB448/oAB809 for nMagHigh1 and pMag insertion, respectively, and for Gibson assembly[62]. The mutation rate was estimated at 1.3-2.9 per kilobase pair through Sanger sequencing. All identified mutations were recloned into the original pAB150 plasmid and used for final characterization. For nMagHigh, the inserts were amplified using primers oAB409/oAB696, and the backbone was amplified from pAB150 with primers oAB589 /oAB446. Before gel purification, the PCR product was digested with DpnI, followed by restriction digestion with AvrII and BglII, gel extraction, and ligation. For pMag, inserts were amplified with primer pair oAB49/oAB285, and the backbone was amplified with primers oAB448/oAB286. Before gel purification, the PCR product was digested with DpnI, followed by restriction digestion with PacI and KpnI-HF, gel extraction, and ligation. Plasmid library sizes were determined based on colony counts of dilution series to 1.2 * 10^6 and 1.9 * 10^6 for pMag and nMagHigh, respectively.

Single point mutations in all mOptoT7 plasmids were based on published plasmids (mOptoT7 Version 1, V1)[58] and created using CloneAmp HiFi PCR Premix (Takara Bio). All constructs were transformed

into Top10 *E. coli* competent cells and checked through sequencing (Microsynth).

## Screening and isolation of variants

We picked the colonies obtained from the sorting into 96-well plates containing M9 medium and grew them overnight to full cell density. We again inoculated main cultures of the individual variants at low cell concentrations so that the cultures are in log growth phase throughout the experiment for single-cell measurements, before inhibition of transcription and translation for maturation of mCherry[8] as follows. Sample was added to an inhibition solution in equal volumes in 96-well U-bottom plates (Thermo Scientific Nunc) on ice, resulting in a final inhibitor concentration of 250 μg/mL rifampicin (Sigma-Aldrich Chemie GmbH) and 25 μg/mL tetracycline (Sigma-Aldrich Chemie GmbH). The inhibition solution contained 500 μg/mL rifampicin and 50 μg/mL tetracycline in phosphate buffered saline (Sigma-Aldrich Chemie GmbH, Dulbecco's phosphate buffered saline) and was filtered using a 0.2 μm syringe filter (Sartorius). After incubated on ice for at least 30 min the samples were transferred to a 37 °C incubator for 90 min for mCherry maturation. Then, the cells were kept on ice until measurement through flow cytometry. From every plate, we then inoculated two 96-well plates and grew them for 5 h either in the dark or in the light before inhibition and measurement through flow cytometry. Based on these first characterizations, we chose a reduced subset of variants with differing mCherry expression levels as well as fold changes higher than the Opto-T7RNAP*(563) "wild-type" (WT) control. These variants were then recharacterized and again a subset of these variants was sent for sequencing. We then re-cloned mutations of unique Magnets variants identified through sequencing into the original Opto-T7RNAP*(563) plasmid. We transformed the recloned variants into AB360 strains containing the pAB50 reporter plasmid, grew them in 96-well plates overnight to full cell density, and then stored the plates in 25% glycerol at -80 °C. For final characterizations, we inoculated main cultures in 96-well plates at low cell concentrations through pin replication (96-pin replicator, Scinomix, MO, USA, SCI-4010-0S) so that the cultures are in log growth phase throughout the experiment before inhibition of transcription and translation for maturation of mCherry through flow cytometry (Fig. 1D, left; for WT).

## Growth and light induction conditions

All experiments were performed in an environmental shaker. The shaking incubator consisted of a Kuhner ES-X shaking module (Adolf Kühner AG, Basel, Switzerland) mounted inside an aluminum housing (Tecan, Maennedorf, Switzerland) and temperature controlled using an "Icecube" (Life Imaging Services, Basel, Switzerland) at 37 °C with shaking at 300 rpm and black, clear bottom 96-well plates (Cell Culture Microplates 96 Well μClear® CELLSTAR®, Greiner Bio-One GmbH, Product #: 655090), which was sealed with peelable foil (Sealing foil, clear peelable for PlateLoc, No. 16985-001, Agilent) to eliminate liquid evaporation and guarantee sterility, as well as a plastic lid (Greiner Bio-One GmbH, Product #: 656171). We used a previously published light induction setup[39]. In this setup, 96 LEDs (SK6812, Dongguang Opsco Optoelectronics) are arranged on a printed circuit board (PCB) at a pitch of 9 mm in an 8 × 12 grid and daisy-chained using their DIN and DOUT ports with a 0.1 nF capacitor placed in parallel with the VDD port of each LED and ordered preassembled (www.pcbway.com). 4 × 96-LED PCBs were powered using a single Adafruit #658 5-V 10-A switching power supply (digikey.ch). The LEDs were controlled by an Arduino Uno microcontroller (Arduino) using the fastLED library (http://fastled.io/). The 96-LED array was mounted inside the shaking incubator using custom three-dimensional (3D)-printed holders (Ultimaker S5 using black Ultimaker co-polyester), and a custom-made anodized aluminum plate (10-mm thick, with 96 holes of 4-mm diameter). A 3D-printed adapter was placed between the aluminum plate and the microtiter plate which were held in place by metal rods (4 mm diameter, 20 mm length). For experiments, cultures were pin replicated into fresh M9 medium containing the respective inducer concentrations as described in the methods section for the screening and isolation of variants. This high dilution ensures that the cells are still in logarithmic growth phase after 5 h, at the end of the experiment[8]. 200 μl of inoculated culture was incubated per well of the 96-well plates. Cells were grown for 5 h before transcription and translation were stopped with rifampicin and tetracycline[8].

## Flow cytometry measurement

Cell fluorescence was characterized using a CytoFlex S flow cytometer (Beckman Colter) equipped with CytExpert 2.1.092 software. mCherry fluorescence was measured with a 561 nm laser and 610/20 nm band pass filter and the following gain settings: forward scatter 100, side scatter 100, mCherry gain 300. Thresholds of 2500 FSC-H and 1000 SSC-H were used for all samples. The flow cytometer was calibrated before each experiment with QC beads (CytoFLEX Daily QC Fluorospheres, Beckman Colter) to ensure comparable fluorescence values across experiments from different days. For the primary high-throughput characterization of the sorted variants, measurements were aborted after 5000 events or after 30 s. As threshold, 15,000 events were recorded in a two-dimensional forward and side scatter gate, which was drawn by eye and corresponded to the experimentally determined size of the testing strain at logarithmic growth and was kept constant for analysis of all experiments and used for calculations of the median and CV using the CytExpert software (Supplementary Fig. 31). At least 50,000 events were taken that were distributed amongst 3-9 biologically independent samples with an average of 27,000–31,000 events/sample, depending on the temperature for bacterial experiments. For mammalian experiments, at least 20,000 events were recorded per sample. Cells were inhibited with rifampicin and tetracycline, and mCherry was matured before measurement[8].

## Spectrophotometric and fluorometric measurements

Cells were grown overnight in a 96-well master plate containing 200 μl of M9 liquid media and the appropriate antibiotics in the dark. This master plate was pin-replicated into 96-well black, clear bottom 96-well plates (Cell Culture Microplates 96 Well μClear® CELLSTAR®, Greiner Bio-One GmbH, Product #: 655090), containing 200 μl of fresh M9 liquid media plus antibiotics and sealed with peelable foil (Sealing foil, clear peelable for PlateLoc, No. 16985-001, Agilent). The 96-well plates contained each strain in technical triplicates. Measurements in a Tecan Infinite 200Pro and Firmware v. 3.40 were multiplexed for absorbance at 600 nm (to measure bacterial growth) and fluorescence at 535 nm (ex) / 595 nm (em) to quantify mCherry expression. The reader settings were adjusted to 9 nm bandwidth, 25 flashes, and 200 ms settle time for absorbance mode and 25 nm bandwidth for excitation, 35 nm bandwidth for emission, gain 100, 25 flashes, integration time 2000 μs, and 200 ms settle time. Plates were incubated in an environmental shaker as described above. Light induction was performed using the 96-LED array described above at intensities ranging from 0 (dark control) to 0.96 W m$^{-2}$ at 30 °C and from 0 (dark control) to 5.77 W m$^{-2}$ at 37 °C and 40 °C. Measurements were taken at 1 h time intervals with the aid of a robotic arm[39]. At 30 °C and 37 °C, we observed overgrowth of the bacterial cultures in some of the wells that were accompanied by higher fluorescence levels out of the range observed within the replicates and in comparison to other variants. These samples were excluded from the analysis using the criterion: absorbance at 600 nm > 0.805. This excluded an average of 31 and 15 samples per experiment, coming up to 5.5% and 2.6% of the total number of samples at 30 °C and 37 °C, respectively. Background subtraction was only done at 40 °C because at lower temperatures the background levels (fluorescent levels of the negative control not expressing mCherry) did not affect the analysis of the data.

### Fluorescence-Activated Cell Sorting (FACS)

10 mL M9 was inoculated with 50 μL of glycerol stocks of the libraries and grown in dark tubes overnight at 37 °C and 230 rpm. AB363 controls were prepared accordingly. From these overnight cultures, 2 μL was used for inoculation of 10 mL of M9 (1:5000 v:v), mixed through vortexing and distributed to single wells of a 96-well plate, which were immediately incubated and exposed to the respective light conditions for 4 hours at 37 °C. The respective libraries or controls were then again pooled and pelleted at 3220 x g (4000 rpm) for 15 min and 4 °C. The supernatant was discarded, and the pellet was resuspended in ice-cold PBS. The samples were kept on ice in dark tubes until they were sorted. It was determined before that mCherry fluorescence values of cells in PBS at 4 °C remain constant. Single-cell sorting was performed using a Sony SH800S Cell Sorter and a 70 μm chip. Fluorescence of mCherry was measured with a 561 nm laser and a 617/30 nm bandpass filter. The sorting chamber was maintained at 4 °C and sample chamber pressure was never set higher than four arbitrary units on the machine. Gain settings were set individually at the respective control samples. Sheath fluid was used as a running buffer, and gates for setting the threshold for sorting were individually set. Their stringency varied and was determined by comparison to the respective control sample. For enrichment, cells were sorted into 8 ml LB medium in black tubes (Black Tubes CELLSTAR® (15 mL), Greiner Bio-One GmbH), while for single-cell sorting, selected events were spotted onto LB agar omnitrays (Nunc™ OmniTray™, Thermo Scientific) supplemented with chloramphenicol and ampicillin. Plates were grown at 37 °C overnight. Colonies were picked in 96-well plates (Microplates 96 well polystyrene, Product #: 655161, Greiner Bio-One GmbH) filled with 200 uL LB medium using sterile toothpicks and incubated overnight at 37 °C and 230 rpm and sealed with gas-permeable membrane (BREATHseal™, Greiner Bio-One GmbH). The next day, 100 μL was pipetted into a new 96-well plate and 50% glycerol (PanReac Applichem) was added into both plates which were then sealed and frozen to -80 °C until use.

### Fitting of light dose-response curves

For fitting of mCherry expression in response to light intensity using least squares regression, we used Prism 9 for MacOS (Version 9.3.1 (#350, GraphPad Software, LLC.).

$$f_{OT7}(x) = b + (t - b)\frac{x^n}{I_{50}^n + x^n} \quad (1)$$

Where in Eq. (1) $f_{OT7}(x)$ describes the gene expression controlled by the respective Opto-T7RNAP variant as a function of light intensity, $x$ represents light intensity, $b$ corresponds to the basal promoter under not activated conditions, $t$ is the maximal promoter expression, $I_{50}$ is the light intensity for half-maximal activation, and $n$ is the Hill coefficient for Opto-T7RNAP. As a weighting method, no weighting was chosen, and each replicate Y value was considered as an individual point.

Values for $b$, $t$, $I_{50}$, $n$, and their lower and upper 95% profile-likelihood confidence limits as well as R squared values, were calculated and confidence bands are shown in the respective plots.

Flow cytometry measurements were performed after 5 h induction during the exponential growth phase. As the time point for the spectrophotometric measurement, we chose 8 h for cells grown at 37 °C and 13 h for cells grown at 30°C. To compensate for the slight shift in OD600 of variant N100L 30°C, and thus slightly later fluorescence peak, we took the 15 h time point only for this variant.

### Fitting of growth curves

For fitting of 600 nm absorbance in response to incubation time using least squares regression, we used Prism 9 for MacOS (Version 9.3.1 (#350, GraphPad Software, LLC.).

$$f_{abs}(x) = b + \frac{(t - b)}{1 + 10^{\log T50 * n}} \quad (2)$$

Where in Eq. (2) $f_{abs}(x)$ describes the absorbance at 600 nm due to cell growth as a function of time, $b$ corresponds to the background 600 nm absorbance, $t$ is the maximal 600 nm absorbance, $T50$ is the time for half-maximal culture density, and $n$ is the Hill slope.

Values for $b$, $t$, $T50$, $n$, and their lower and upper 95% profile-likelihood confidence limits as well as R squared values, were calculated.

### Cell culture and Transfection

HEK293T cells (ATCC, strain number CRL-3216) were cultured in Dulbecco's modified Eagle medium (DMEM, Gibco) supplemented with 10% FBS (Sigma-Aldrich), 1% penicillin/streptomycin, 1× GlutaMAX (Gibco) and 1 mM Sodium Pyruvate (Gibco). Cells were kept at 37 °C and 5% $CO_2$[58]. Transfections were carried out in suspension using 24 well plates (either black for light experiments, PerkinElmer, or transparent for dark control, ThermoFisher) at a density of $1.5 \times 10^5$ cells/well; DNA and Polyethylenimine (PEI) (Mw 40 000; Polysciences, Inc.) complexes were incubated for 25 min at room temperature before addition to the cells.

### Light induction and Flow Cytometry analysis of HEK293T cells

Cells were illuminated with constant light using 470 nm LEDs (Super Bright LEDs Inc) in an optimized version of the Light Plate Apparatus (LPA)[58]. HEK293T cells were analyzed 24 h after illumination using CytoFLEX S Flow Cytometer (Beckman Colter)[58]. Measurement was done using 488 and 561 lasers with 530/11 nm and 610/20 nm OD1 bandpass filters, respectively. Cells were washed once with DPBS (Thermo Fisher) and incubated with 100 μL of Accutase solution (Sigma-Aldrich) to allow detachment prior to measurement. FCS/SSC parameters were used to select the main cell population and singlet fraction. For each sample, light-induced expression was normalized to the mCitrine constitutive plasmid (dividing the mRuby3 reporter fluorescence values by the mCitrine fluorescence values) to account for transfection efficiency. Data was analyzed using Cytoflow Software.

### Clustering analysis

Principal component analysis and hierarchical clustering were performed in R using the following packages: cluster[63], factoextra[64], dendextend[65], and ggplot2[66]. The optimal number of clusters was determined using the silhouette algorithm[53] implemented in the factoextra package of R.

The flow cytometry (single cell) data were scaled before PC analysis. We observed that scaling the data before PC analysis for the fluorometric (population) data gave too much weight to the measurements in the lag phase of growth when the mCherry activity was below the detection level of the plate reader due to the low number of cells. Therefore, the fluorometric data were not scaled before PC analysis.

Hierarchical clustering was performed using the Ward method with Euclidean distances[50]. When we analyzed all the data at the three different temperatures together (in both cases: single cell-flow cytometry and population-fluorometric data), we employed the principal components scores to perform the hierarchical clustering rather than the raw data to have the same mean in the data at all temperatures and again to avoid giving special weight to the data at 30 °C, where the mCherry showed the highest values.

### Protein structure visualization

Protein structures were visualized using the PyMOL Molecular Graphics System, Version 2.5.2 Schrödinger, LLC.

## Reporting summary

Further information on research design is available in the Nature Portfolio Reporting Summary linked to this article.

## Data availability

Source data are provided with this paper and are available at Zenodo [https://doi.org/10.5281/zenodo.18815874]. Previously published protein structures are available at the PDB under accession code PDB 3RH8. Source data are provided with this paper.

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

## Acknowledgements
We thank Dr. Tsvetan Kardashliev for helpful discussions and Dr. Luzius Pestalozzi for the testing and the supply of the polymerase and buffer used for error-prone PCR. We further thank Dr. Stephanie Aoki for helpful discussions. We thank the Single Cell and Lab Automation Facility of the DBSSE, ETH Zurich, in particular Dr. Gregor Schmidt, Dr. Aleksandra Gumienny, and Dr. Mariangela Di Tacchio for their excellent support throughout the project. This article is dedicated to the memory of Josep (Pepe) Casadesús. We acknowledge funding from FET-Open research and innovation actions grant under the European Union's Horizon 2020 research and innovation program (CyGenTiG; grant agreement 801041) to M.K. D.C. was a recipient of an EMBO Short-Term Fellowship (Grant number 8903).

## Author contributions
A.B. conceived, planned, and coordinated the project and wrote the manuscript with contributions from all authors. A.B. and Y.W. generated the libraries and performed the FACS. A.B., Y.W., and D.C. designed and performed bacterial experiments and analyzed the corresponding data. D.C. performed the PC analysis and hierarchical clustering. S.D. performed experiments in mammalian cells and analyzed the corresponding data. M.K. supervised the project and provided funding.

## Funding

## Competing interests
The authors declare no competing interests.
