## [Transparent Peer Review file · Nature Communications]

Enhancing the performance of Magnets photosensors

Corresponding Author: Professor Mustafa Khammash

Version 0:

Reviewer comments:

Reviewer #1

(Remarks to the Author)

In this paper, Baumschlager et al applied random mutagenesis followed by high-throughput screening in both dark and light conditions on the nMagHigh1 and pMag in the context of opto-T7RNAP with the goal of improving their light sensitivity. They identified many hits with altered mCherry expression compared to the original Magnet photoreceptors, and proceeded to characterize these changes at three different temperature and over time, with a final brief look at T7-based gene expression in mammalian cells.

In general, I think the author's idea on improving light sensitivity through mutagenesis and characterization to be interesting. Among the issues facing optogenetic tool application in mammalian contexts, light toxicity and inefficient tissue penetration could indeed be solved through improvement of light sensitivity. However, what this paper has in breadth it lacks in depth – we are left with very little idea of what exactly is changed in many of these mutants, nor do the authors leave us with a fully-characterized, improved tool to use in future experiments.

Major concerns:

- Most broadly, the concept of "light sensitivity" here is poorly defined, and an amalgamation of many different possible biological processes. A higher mCherry expression level at a particular light intensity could be due to any of following: (1) increased expression levels of the split-T7 components, (2) slower reversion kinetics from the lit to dark state, (3) a tighter binding affinity between the Magnet components, leading to a higher concentration of reconstituted T7s at lower light intensities, or (4) an actual change in the absorbance of photons by individual Magnet photoreceptors. To fully understand the new mutants, it is important to have some idea whether they are simply expressed at higher levels in bacteria (something that may not translate to other contexts), or if they maintain the photoactivated state for longer (something that would substantially affect experimental design, such as the use of pulsatile inputs), or if they are simply a higher-affinity binding pair.

The manuscript would be greatly improved if the authors would fully characterize at least one improved Magnet variant (e.g. the one with the most improved light sensitivity) to understand exactly what is changing, and rule out the other mechanisms. Many of these experiments would be quite straightforward to do: the levels of T7-Magnet fusions (or a T7-Magnet-GFP variant, say) can be measured to test for changes in expression, and dark state reversion kinetics can be measured spectroscopically, by measuring Magnet association/dissociation after a change from light to darkness, or potentially by looking at mCherry decay kinetics after a switch to darkness.

The authors appear to argue that any such mechanistic insights are "out of the scope of our engineering-driven approach" (p. 18) but this is a pretty extreme position. Mutagenesis experiments have been carried out by biologists for a half century (error-prone PCR is itself at least 30 years old), yet here the authors appear to deem these approaches "engineering approaches" and absolve themselves of any subsequent mechanistic studies. This is in stark contrast to excellent screening & characterization work done on LOV domains over the past decade (e.g., Guntas et al, PNAS) who isolated mutants, measured changes in binding affinity, and characterized binding kinetics in mammalian cells. As it stands, any user who wishes to use an improved Opto-T7 (or other Magnet system) based on these results would have no idea which mutation to use for a particular application, nor how it works!

- A related issue: the authors did many experiments in bacterial cells where light sensitivity is not typically a major issue, but did not measure the light sensitivity in the mammalian cell context where it is likely to be more important. In mammalian cells, only one light intensity condition was tested. An investigation of light sensitivity in this context would help the reader to

know whether the light sensitivity property screened for in bacteria could be easily translated to other model systems.

Minor concerns:

- The title is misleading – “direct evolution” refers to iterative rounds of mutagenesis and selection. Here the authors don’t carry out any iterative process – just a single round of mutagenesis followed by screening.
- It seems that the authors only focused on mutants either in nMag or in pMag. Have the authors ever tried combining the mutants with increased light sensitivity in nMag and pMag together and see if they could have synergistic effect?
- It appears that opto-T7-induced expression is relatively weak in mammalian cells, in comparison to excellent Gal4-based gene expression systems (e.g. GAVPO from Wang et al Nature Methods). The authors might consider applying their improved Magnets in a mammalian context where they have already been used more effectively, such as split Cas9 (Nihongaki et al., 2015).

Reviewer #2

(Remarks to the Author)

The authors use directed evolution and a high-throughput screening strategy to identify new photosensitive Magnet domains with altered light-activity dose-response curves. In this approach the nMag and pMag components are optimized separately but some common amino acids (T69 and N100) that alter photosensitivity as well as the basal dark and the maximal light induced expression are identified. Optogenetic tools with significantly higher photosensitivity would certainly be of value in decreasing light toxicity and opening the possibility of multiplexing. Yet, the here obtained new Magnets only have slightly altered light sensitivity of ca. 50% and the overlap of the response windows with the original Magnets is still very high. Therefore, multiplexing does not seem feasible with the new tool and is also not shown. Some of the mutations result in improvements in Magnets in terms light response and thermal stability but these are shadowed by already reported enhanced magnets (eMag) (Benetti et al. eLife 2020;9:e63230. DOI: <https://doi.org/10.7554/eLife.63230>). Actually, some of the mutations found here in the high-throughput screen overlap with the eMag system. At present this work feels better suited to a more specialized journal and does not open completely new perspectives in optogenetics that would justify the publication in Nature Communications. Below are some suggestions for improvement of the manuscript.

- 1) The Opto-T7RNAP tool requires both the pMag and nMagHigh1 component to be photoactivated. Therefore, the increased light sensitivity of one component might be limited by the less efficient activation of the second one. Beneficial mutations found in one component should be tested in the other one, considering that the here described library certainly doesn’t cover the entire sequence space and that the photochemistry in the pMag and nMagHigh1 are the same.
- 2) In the current study, one component is optimized at a time and the other one is kept constant. While this certainly is sensible in the initial high throughput screen, the combination of the best performing pMag and nMagHigh1 variants should be tested to investigate if the benefits are additive or not.
- 3) In the study of saturating mutagenesis at the hotspots T69 and N100 the terminology of the manuscript changes and fold mCherry under high intensity blue light are reported and not the previously used parameters b , t and I_{50} . The term light sensitivity should be used only in one sense: either the light intensity required to induce half-maximal gene expression or the light intensity used to obtain a certain gene expression. It is misleading that these differences in mCherry expression are set equal to higher light sensitivity as they might simply arise from a higher overall expression under blue light. For the most promising mutations the full characterization including dark activity and full light sensitivity profiles should be measured.
- 4) The thermal stability is an obvious limitation of the Magnets. This begs the question if the increased light sensitivity originates from a higher fraction of correctly folded proteins in the cells or is it an intrinsic property of the protein that can be explained through its photochemistry. Spectroscopic characterization with purified proteins and molecular modeling would help answer these questions.
- 5) In many cases the increased maximal light induced expression comes at the cost of higher basal expression in the dark. A ratio of blue light to dark expression should be added as an additional parameter of merit.
- 6) For the expression in mammalian cells the improvements remain minor and the increased fold expression under light illumination does not necessarily show increased light sensitivity of the protein itself but might just be due to more efficient dimerization between the nMag and pMag fragment or improved thermal stability. A full light sensitivity profile would be required to support this claim. A light intensity of 0.01 mW cm^{-2} is set as sub-saturating light induction but at what intensity does the system saturate and what is the range that the expression can be tuned over? These values could be rather different in *E.coli* and mammalian cells.

Minor points:

- 1) The library coverage of the high-throughput screen should be mentioned and discussed in the manuscript.
- 2) There are some inconsistencies in the units of light intensity. On page 5 the high light intensity is given as 3.85 W/cm^2 , in the Figures 1-3 the intensity is in up to 6 W/m^2 and in Figure 4 the intensities are in mW/cm^2 . These numbers differ orders of magnitude and should be verified.

Version 1:

Reviewer comments:

Reviewer #1

(Remarks to the Author)

I thank the authors for their response to my prior review and have a few additional comments:

(1) First and foremost, I still not convinced that this manuscript is a strong candidate for Nature Communications. The authors argue that their paper is principally focused on tuning a synthetic biology gene expression system, not understanding mechanism, which is fine. However, it narrows the scope of the paper considerably - we might learn how OptoT7 in bacteria change, but we do not gain much insight in how these changes will translate to any other application, since we are left with little mechanistic insight. We also are not left with a demonstration that the higher light sensitivity obtained from the screen is useful, say for example by regulating metabolism in high-OD culture. I agree with the other reviewer's first review that this manuscript would be better suited for a more specialized journal.

(2) In their rebuttal the authors reference some supplementary data from their prior paper (Baumschlager 2017). In my view, that paper has a wealth of additional results that support the premise of the current study, and it would be *very* helpful if they were explained in the introduction of this study. The 2017 paper investigates two classic VVD mutants, I85V and I74V/I85V, which have well known effects on photocycling time, but observe that one of these mutants exhibits a massive change in light sensitivity (Fig 4 and S1 1), whereas the other does not. This shows that sensitivity and photocycle time can be tuned independently - they are not one-to-one - and that wild-type VVD is far from optimal in sensitivity. I agree with the authors' rebuttal that these results argue against sensitivity arising simply from variation in protein expression level. They also argue against the sensitivity being solely a function of lit-state lifetime.

The manuscript would be strengthened if the authors would describe something like this logic in their current manuscript, because it is absolutely not clear simply by reading this paper whether light sensitivity itself is an engineerable parameter, since so many other variables could in principle influence a dose response curve.

This prior work also suggests that the authors have already identified a very sensitive variant in I74V/I85V. Why was this variant not included in the current analyses, at least as a positive control? Would it be competitive with the best (most sensitive) variants discovered here?

(3) The authors' table detailing the sensitivity changes (Supp Table 13) is very useful. It is one of the key pieces of information for users who might be interested in implementing these mutations for further study. I would strongly prefer that this be included in the main text as a final figure panel or table, not on page 38 of the supplementary information where it is likely to escape notice by most readers.

(4) I have a major concern about the quantification and analysis of the mammalian cell data in Figure 3. It is quite misleading to only compare fold-changes between mutants in the light to the wild-type variant in the dark. This is because many variants already show dramatically increased gene expression in the dark! Fold-changes should be reported for each variant between its dark and light conditions. Separately, the authors can describe changes in "leakiness" in the dark, for example showing that M48V N100L is already ~10-fold higher than wild-type in the dark. It is not at all clear that the changes between dark and low-intensity light are significant for many variants (e.g. N100L or M48V/N100L). Is the fold-change between dark and 0.1 W/m² statistically different for the mutants compared to the wild-type in mammalian cells? Without such a comparison it is not clear that the changes in sensitivity extend to the mammalian context at all.

Reviewer #2

(Remarks to the Author)

We have revised our manuscript according to the reviewers' comments. Reviewer comments are in black, our responses have blue font. All changes in the manuscript and supplementary material have font color blue.

Reviewer #1 (Remarks to the Author):

In this paper, Baumschlager et al applied random mutagenesis followed by high-throughput screening in both dark and light conditions on the nMagHigh1 and pMag in the context of opto-T7RNAP with the goal of improving their light sensitivity. They identified many hits with altered mCherry expression compared to the original Magnet photoreceptors, and proceeded to characterize these changes at three different temperature and over time, with a final brief look at T7-based gene expression in mammalian cells.

In general, I think the author's idea on improving light sensitivity through mutagenesis and characterization to be interesting. Among the issues facing optogenetic tool application in mammalian contexts, light toxicity and inefficient tissue penetration could indeed be solved through improvement of light sensitivity. However, what this paper has in breadth it lacks in depth – we are left with very little idea of what exactly is changed in many of these mutants, nor do the authors leave us with a fully-characterized, improved tool to use in future experiments.

Major concerns:

- Most broadly, the concept of “light sensitivity” here is poorly defined, and an amalgamation of many different possible biological processes. A higher mCherry expression level at a particular light intensity could be due to any of following: (1) increased expression levels of the split-T7 components, (2) slower reversion kinetics from the lit to dark state, (3) a tighter binding affinity between the Magnet components, leading to a higher concentration of reconstituted T7s at lower light intensities, or (4) an actual change in the absorbance of photons by individual Magnet photoreceptors. To fully understand the new mutants, it is important to have some idea whether they are simply expressed at higher levels in bacteria (something that may not translate to other contexts), or if they maintain the photoactivated state for longer (something that would substantially affect experimental design, such as the use of pulsatile inputs), or if they are simply a higher-affinity binding pair.

We agree that “light sensitivity” is a generic term that we use to describe a change in the observed phenotype outcome (increased or decreased expression of the reporter protein) in response to different constant intensities of light. We used different experimental approaches for our in vivo characterization approach, to disentangle different effects that might lead to this observed phenotype (e.g., reversion kinetics, temperature stability).

Changes in the manuscript:

While we kept the overall term “light sensitivity” to refer to this general phenotype, we understand the point of the reviewer and now defined the term and described the different properties in the results and discussion section. We now also clearly separate the two directly observable properties of our characterization: changes in I50 values, to which we now refer to as “light sensitivity” and changes in basal and maximal expression to which we now refer to as increased or decreased “light activation”. We added an additional description to clarify this and changed the terminology throughout the manuscript. Regarding the expression levels of the Opto-T7RNAP parts, please see our response to the next comment. We agree that in our in vivo characterization, kinetics were missing. As the reviewer correctly mentions, this can substantially affect experimental design. We thus performed further experiments to complement our previous characterization. Please also refer to our response to the following comment, which, in part, resolves around the same question.

The manuscript would be greatly improved if the authors would fully characterize at least one improved Magnet variant (e.g. the one with the most improved light sensitivity) to understand exactly what is changing, and rule out the other mechanisms. Many of these experiments would be quite straightforward to do: the levels of T7-Magnet fusions (or a T7-Magnet-GFP variant, say) can be measured to test for changes in expression, and dark state reversion kinetics can be measured spectroscopically, by measuring Magnet association/dissociation after a change from light to darkness, or potentially by looking at mCherry decay kinetics after a switch to darkness.

The authors appear to argue that any such mechanistic Insights are “out of the scope of our engineering-driven approach” (p. 18) but this is a pretty extreme position. Mutagenesis experiments have been carried out by biologists for a half century (error-prone PCR is itself at least 30 years old), yet here the authors appear to deem these approaches “engineering approaches” and absolve themselves of any subsequent mechanistic studies. This is in stark contrast to excellent screening & characterization work done on LOV domains over the past decade (e.g., Guntas et al, PNAS) who isolated mutants, measured changes in binding affinity, and characterized binding kinetics in mammalian cells. As it stands, any user who wishes to use an improved Opto-T7 (or other Magnet system) based on these results would have no idea which mutation to use for a particular application, nor how it works!

We want to respond to the two paragraphs above in a single response, as we think that they contain arguments that are connected. We understand, and it is mentioned in the manuscript, that mechanistic comprehension would

add an interesting angle to the manuscript. In our opinion, in the excellent article by Guntas et al at PNAS (2015) authors performed a screening for mutants with enhanced dimerization properties, and then characterized these mutations from a biochemical perspective, therefore providing a very nice focus on the role of the PAS domain residues in dimerization. Applications of the dimerization enhanced variants are also shown in the article. However, the aim of this manuscript is to provide an engineering workflow for the fast tuning of optogenetic protein with the application in regulatable expression tools for synthetic biological applications. Therefore, we focused on developing a very different tool from the previous articles, and consequently our study required a different set of experiments. We thus performed an extensive set of characterization experiments, which cover a very wide range of application scenarios and important parameters all related to the regulation of gene expression. Our focus is on the *in vivo* behavior of the mutant variants to provide a complete portfolio of tools for different applications. For this, we combined both time-resolved expression data that covers all growth phases, with single-cell expression data to further investigate expression heterogeneity. These clearly support and demonstrate our goal to tune the properties of the proteins toward desired features. This follows a different objective than the wonderful work of mentioned publications and the mechanistic insights they generated. The main objective was to enable the tuning of these optogenetic tools and thus enable a broader use in applications and not mechanistic investigations of photochemistry, which we better leave to specialized groups. While such analysis is highly interesting on a theoretical level, we provided a large dataset that goes beyond most experimental sets in protein engineering. By characterizing 1) the behavior of the variants experimental over time at the population level, we provide crucial information for use in bioprocesses and bulk experiments. Such information cannot be extracted from mechanistic studies, but could be essential for practical applications. Users can infer from these data how the variants behave from the inoculation point to stationary phase, all at different conditions (e.g., light intensities or temperatures) and in addition have information about population heterogeneity. To obtain this characterization, we had to develop an automation setup that transfers the samples between the light induction device and the measurement device. 2) In addition to these spectrophotometry measurements, which would already be a very good characterization of the newly generated variants, we further investigated changes in heterogeneity on the single-cell level through flow cytometry, important not only for targeted induction but also to more precisely control populations of cells to desired outputs. 3) On top of this, we tested altered behavior under different temperatures, again with both methods.

Changes in the manuscript:

- 1) We now further expanded this extensive *in vivo* characterization and added a set of practical experiments for on-off kinetics and thus not only provide a wide set of variants that show different behavior but also created a comprehensive experimental dataset that covers all relevant aspects for the application of these variants in biotechnological and synthetic biology contexts.
- 2) We also thank this reviewer for bringing up how a future user can decide which variant to use for a particular application. To further assist users in choosing the appropriate mutation for their specific applications, we have now included an overview table classifying the different variants according to their properties. We think that this makes applying the results of this work easier for other researcher and thus thank the reviewer for this suggestion.

We also want to clarify specifically why changes in internal protein concentrations cannot explain the altered behavior of the variants we identified. We agree that differences in the concentration of transcriptional regulators can lead to changes in the resulting expression level. Mutations might cause slight differences in the expression levels of a protein. However, we did not quantify potential slight changes in the translation of the regulator that are caused by the mutations in our particular case, because of the following reasons:

- 1) Light sensitivity changes (in terms of I50) cannot be attributed to changes in expression levels of Opto-T7 parts:
During the development of the Opto-T7RNAPs, we tested different expression levels of the two split parts. Even though the changes in the expression levels of the regulator parts led to dramatic changes in the reporter gene expression (compare Figure 2 BC of the publication Baumschlager et al. ACS Synth Biol 2017), the dose-response curves were not changed due to the altered regulator expression levels. This is highly comparable to potential changes caused by altered translation rates, especially because also there, the expression level of one of the split parts in relation to the other part would be altered. (Figure S11 “Light dose-response curve is not changed by different domain expression levels” of the publication Baumschlager et al. ACS Synth Biol 2017). We even tested this for different Magnet versions as shown in this figure (Opto-T7RNAP*(563-F1) and Opto-T7RNAP*(563-F2)) In case there would be changes in the expression level caused by a mutation, this would also only alter the expression level of one of the Opto-T7RNAP parts, as simulated in these experiments. Thus, changes in light sensitivity cannot solely be attributed to expression level changes of the Opto-T7RNAP parts.
- 2) Large changes in Opto-T7RNAP expression are required to lead to moderate changes in the resulting Opto-T7RNAP transcription, which we previously determined using an arabinose-inducible promoter. This promoter is known for large differences in uninduced vs arabinose-induced transcription. From leaky expression of the promoter to full induction for expression of the Opto-T7RNAP parts, which means a

dramatic change in the concentration of both Opto-T7RNAP elements (usually this promoter increases protein of interest concentrations by several hundred-fold), led to a 4-fold increase in the resulting reporter gene expression comparing both dark and light-induced conditions separately (Figure 4 of the publication Baumschlager et al. ACS Synth Biol 2017). In addition, we observed that only relatively high arabinose induction (more than 0.05% arabinose) leads to significant changes in the resulting output (same figure). Thus, we think it is very unlikely that such large differences of expression can be caused by a single or double amino acid substitution. In addition, in the variants presented in this manuscript, we observed fold changes of up to 10-fold compared to the wild-type. This leads us to conclude that other properties than expression level changes need to be the root cause of the observed phenotypes of the identified variants. We thus focused on protein stability, and in particular temperature stability, as the reviewer correctly mentions that mutations might lead to increased stability of the protein. Indeed, what we found during this characterization is that some variants, such as N100L, show improved temperature stability.

Changes in the manuscript:

We agree with the reviewer that association/dissociation can also play an important role in the observed light sensitivity. We thus conducted further experiments to characterize the mCherry decay kinetics as suggested by the reviewer. We thank the reviewer for motivating us to these additional experiments as they are valuable contributions for implementing these mutations for different purposes.

- A related issue: the authors did many experiments in bacterial cells where light sensitivity is not typically a major issue, but did not measure the light sensitivity in the mammalian cell context where it is likely to be more important. In mammalian cells, only one light intensity condition was tested. An investigation of light sensitivity in this context would help the reader to know whether the light sensitivity property screened for in bacteria could be easily translated to other model systems.

It is true that the focus was to develop and test the variants for use in bacteria. However, we show that – in principle – these mutations could also be highly interesting in mammalian cells.

Changes in the manuscript:

To make this suggestion even stronger, we followed the advice of the reviewer and added additional experiments in mammalian cells. We performed experiments in the dark, at a sub-saturating light induction condition (0.1 W m⁻²) and close to saturating regulator activity (0.8 W m⁻²). Through the additional characterization conditions, we could further boost the activity of the mOptoT7 to about 50-fold compared to the WT, whereas the WT control increased by 14-fold. We thank the reviewer for suggesting these further experiments as they significantly improved our manuscript.

Minor concerns:

- The title is misleading – “direct evolution” refers to iterative rounds of mutagenesis and selection. Here the authors don’t carry out any iterative process – just a single round of mutagenesis followed by screening.

Changes in the manuscript:

We changed the title to “Enhancing the performance of Magnets photosensors”.

- It seems that the authors only focused on mutants either in nMag or in pMag. Have the authors ever tried combining the mutants with increased light sensitivity in nMag and pMag together and see if they could have synergistic effect?

We observed synergistic effects within the individual components. Some of the mutations were characterized in tandem with a second mutation. Of course, also combinations of different mutations in the different dimerization domains might lead to further improvements and further possibilities for tuning.

Changes in the manuscript:

We expanded on this in the discussion section to provide an overview of our findings and maybe also inspire further research as combinatorial assays enable extremely large sets of investigations.

- It appears that opto-T7-induced expression is relatively weak in mammalian cells, in comparison to excellent Gal4-based gene expression systems (e.g. GAVPO from Wang et al Nature Methods). The authors might consider applying their improved Magnets in a mammalian context where they have already been used more effectively, such as split Cas9 (Nihongaki et al., 2015).

We agree with this reviewer and acknowledge that this reviewer appreciates that there are numerous potential applications for the mutations discovered in this work. The mOptoT7 is just one of these. We also believe that these

suggestions would be a fantastic follow-up after this work gets published, and we are looking forward to seeing our results applied to these systems. For now, it is out of the scope of this study.

Changes in the manuscript:

With new experiments added to the manuscript, we tackle exactly one point of the reviewer, which is the weak expression of mOptoT7 in mammalian cells. As previously mentioned, we could boost the activity of the mOptoT7 to about 50-fold compared to the WT, successfully addressing this issue.

We thank the reviewer for their insightful comments and suggestions which significantly improved the manuscript.

Reviewer #2 (Remarks to the Author):

The authors use directed evolution and a high-throughput screening strategy to identify new photosensitive Magnet domains with altered light-activity dose-response curves. In this approach the nMag and pMag components are optimized separately but some common amino acids (T69 and N100) that alter photosensitivity as well as the basal dark and the maximal light induced expression are identified. Optogenetic tools with significantly higher photosensitivity would certainly be of value in decreasing light toxicity and opening the possibility of multiplexing. Yet, the here obtained new Magnets only have slightly altered light sensitivity of ca. 50% and the overlap of the response windows with the original Magnets is still very high. Therefore, multiplexing does not seem feasible with the new tool and is also not shown. Some of the mutations result in improvements in Magnets in terms light response and thermal stability but these are shadowed by already reported enhanced magnets (eMag) (Benetti et al. eLife 2020;9:e63230. DOI: <https://doi.org/10.7554/eLife.63230>). Actually, some of the mutations found here in the high-throughput screen overlap with the eMag system. At present this work feels better suited to a more specialized journal and does not open completely new perspectives in optogenetics that would justify the publication in Nature Communications. Below are some suggestions for improvement of the manuscript.

We are delighted about the shared opinion of the reviewer that higher photosensitivity of optogenetic tools is of high value for biological research. We disagree on the assessment of novelty, as we believe that the fact that some of the mutations were already published validates our study. Benetti et al eLife (2020) employed a rationale design approach and four out of the eight residues that they identified were also found in our screening. However, by using random mutagenesis and a selection screening we identified different mutations in those residues. Furthermore, our saturation mutagenesis of two residues allowed us to identify the best variants in those residues, something that is very difficult to achieve by pure rational design. Our work also demonstrates the feasibility and power of directed evolution to enhance the properties of current syn bio tools. The light sensitivity of the wild-type Opto-T7RNA is up to two-fold compared to the variants found in this study. Furthermore, these variants showed up to 5-fold increase in the maximum activity and more importantly, up to 5-fold increase in the fold change at high temperature. The study by Benetti et al eLife (2020) focused on the application of the Magnets for dimerization of proteins and the consequent regulation of protein recruitment and inter-organellar contacts. This work offers an alternative new approach for photosensor tuning with a completely different focus: the regulation of gene expression.

1) The Opto-T7RNAP tool requires both the pMag and nMagHigh1 component to be photoactivated. Therefore, the increased light sensitivity of one component might be limited by the less efficient activation of the second one. Beneficial mutations found in one component should be tested in the other one, considering that the here described library certainly doesn't cover the entire sequence space and that the photochemistry in the pMag and nMagHigh1 are the same.

Indeed, it can be expected that some mutations will be easily transferable between the two domains. This is the case for mutation N100L, which we characterized in both domains separately.

Changes in the manuscript:

We expanded on the transferability of the mutations between the domains to make future users of the method and the mutations aware of the possibility of transferability between the domains and combinatorial assays and applications. We also created an overview table (Supplementary Table 13) to give the reader a better overview of the findings and potential expansions. Please also see the corresponding response to the comment of reviewer 1.

2) In the current study, one component is optimized at a time and the other one is kept constant. While this certainly is sensible in the initial high throughput screen, the combination of the best performing pMag and nMagHigh1 variants should be tested to investigate if the benefits are additive or not.

We did observe synergistic effects within the individual components. Some of the mutations were characterized in tandem with a second mutation. Since the different variants have different specific properties, it is not clear what the best set of mutations would be as the choice will be application-dependent, and combining all mutations would not be feasible and out of scope for this study, but would be a nice follow-up.

Changes in the manuscript:

We describe the double mutants as examples of these, where we could entangle these effects further in the text. We further elaborated on these points to make the reader aware of these valid points. Please also see our response to reviewer 1.

3) In the study of saturating mutagenesis at the hotspots T69 and N100 the terminology of the manuscript changes and fold mCherry under high intensity blue light are reported and not the previously used parameters b , t and I_{50} . The term light sensitivity should be used only in one sense: either the light intensity required to induce half-maximal gene expression or the light intensity used to obtain a certain gene expression. It is misleading that these differences in mCherry expression are set equal to higher light sensitivity as they might simply arise from a higher overall expression under blue light. For the most promising mutations the full characterization including dark activity and full light sensitivity profiles should be measured.

It is true that we did not use parameters b , t , and I_{50} for comparison. This is due to the lower number of light intensities (only two, low and high), which does not allow for fitting our dose-response model, from which these parameters are derived from. Due to the large number of variants for each library, it is not feasible to perform a full dose curve for each variant. Also, this was never the goal. Our goal was to investigate the two main hotspots we identified and see if variants with even higher light sensitivity or light activation can be identified, which might not have been accessible via single nucleotide exchanges. For this, a saturation light induction (to identify increased light activation) and a sub-saturating light intensity (to variants with a higher light sensitivity) were sufficient. Although we did not identify variants with improved properties compared to our previously identified ones, this was a good confirmation that we are not missing other even more beneficial variants. As suggested by the reviewer, the -by far- most promising variants were characterized with full light sensitivity profiles (T69M, N100L).

Changes in the manuscript:

We clarified the above-mentioned points in the text of the corresponding paragraph. We thank the reviewer for making us aware that this was not clearly communicated before.

4) The thermal stability is an obvious limitation of the Magnets. This begs the question if the increased light sensitivity originates from a higher fraction of correctly folded proteins in the cells or is it an intrinsic property of the protein that can be explained through its photochemistry. Spectroscopic characterization with purified proteins and molecular modeling would help answer these questions.

We agree that different effects can lead to the same phenotype, so we characterized the ones that are most relevant for experimental design (light intensity, temperature stability) and we further strengthened these characterizations with new experiments of temporal dynamics. We agree that *in vitro* characterization of the protein variants would be very interesting to further elucidate the exact mechanisms of the observed phenotypes, and this is a very nice suggestion for a follow-up of this work. The idea for the follow-up study could be the complete biochemical characterization of some selected variants to decipher the molecular consequences of the mutations. In this work, we focused on changes toward a desired phenotype, and characterized the behavior of the variants under different conditions to provide a complete portfolio of variants for different applications.

Changes in the manuscript:

We have addressed the question on the molecular mechanism experimentally by 1) providing additional *in vivo* characterization of important variants (especially in response to stability and the resulting dynamics as described in the new section "Dynamic off-switching at 37°C"), and 2) a detailed explanation on the effect of Opto-T7RNAP concentration changes in our response to reviewer 1, comment 2.

5) In many cases the increased maximal light induced expression comes at the cost of higher basal expression in the dark. A ratio of blue light to dark expression should be added as an additional parameter of merit.

Changes in the manuscript:

As suggested, we added both the fold change as well as the comparison to the wild type to all the data tables in the supplementary information. Thank you for the comment.

6) For the expression in mammalian cells the improvements remain minor and the increased fold expression under light illumination does not necessarily show increased light sensitivity of the protein itself but might just be due to more efficient dimerization between the nMag and pMag fragment or improved thermal stability. A full light sensitivity profile would be required to support this claim. A light intensity of 0.01 mW cm⁻² is set as sub-saturating light induction but at what intensity does the system saturate and what is the range that the expression can be tuned over? These values could be rather different in *E. coli* and mammalian cells.

Indeed, as the reviewer correctly mentions, these values are different in *E.coli* and mammalian cells. We chose 0.01 mW cm⁻² for showing changes in light sensitivity, as this value was determined to be sub-saturating (see Figure 1d of publication Dionisi et al., ACS Synthetic Biology, 2022 and Figure 4 c of the same publication).

Changes in the manuscript:

We have added this missing explanation in the manuscript. We have further added additional experiments in mammalian cells to expand the characterization. These experiments now show dark, sub-saturating and close-to-saturating light induction of the regulator. At the additional conditions, we could further boost the activity of the mOptoT7 to about 50-fold compared to the WT, whereas the WT control increased by 14-fold. We thank the reviewer for guiding us to these additional experiments.

Minor points:

1) The library coverage of the high-throughput screen should be mentioned and discussed in the manuscript.

Changes in the manuscript:

We added this information to the methods section "Plasmids and genetic parts".

2) There are some inconsistencies in the units of light intensity. On page 5 the high light intensity is given as 3.85 W/cm², in the Figures 1-3 the intensity is in up to 6 W/m² and in Figure 4 the intensities are in mW/cm². These numbers differ orders of magnitude and should be verified.

Thank you for spotting this!

Changes in the manuscript:

We have corrected this and harmonized the units.

We express our sincere gratitude to the reviewer for their insightful review and invaluable suggestions that have contributed to our manuscript.

Point-by-point response to the reviewers' comments.

We have addressed all points raised by reviewer #1. Reviewers' comments are in blue font and the response to the comments is shown in black font. All changes in the main manuscript and supplementary material are shown in green font.

Reviewer #1 (Remarks to the Author):

I thank the authors for their response to my prior review and have a few additional comments:

- (1) First and foremost, I still not convinced that this manuscript is a strong candidate for Nature Communications. The authors argue that their paper is principally focused on tuning a synthetic biology gene expression system, not understanding mechanism, which is fine. However, it narrows the scope of the paper considerably - we might learn how OptoT7 in bacteria change, but we do not gain much insight in how these changes will translate to any other application, since we are left with little mechanistic insight. We also are not left with a demonstration that the higher light sensitivity obtained from the screen is useful, say for example by regulating metabolism in high-OD culture. I agree with the other reviewer's first review that this manuscript would be better suited for a more specialized journal.

We want to emphasize that this manuscript not only provides a novel set of optogenetic tools covering a wide range of light sensitivities and expression levels, but also reports a framework for any research or application that requires tuning an optogenetic tool or expression system. It is a general methodology that is not limited to a specific application. By thoroughly characterizing the effect on protein expression, we demonstrate the applicability for control of gene expression. One of the main points that this work shows is that the most important features of an expression system, i.e., the sensitivity and the expression levels can be tuned to fit specific desired requirements. Our pre-defined goal was to achieve a higher gene expression using the Opto-T7RNAP at lower light intensities, which was achieved as shown by the data. The data presented in this manuscript have been confirmed using two different technologies (flow cytometry and fluorescent plate reading), both of which are in perfect agreement, and cross-validating the findings.

To address the comment about the behavior of the variants at different ODs, we added the full time courses of optical density, reaching from lag to stationary growth phase, together with the reporter expression data and further analysis, providing full information on the expression levels at any stage (and OD) during the entire growth curve, including stationary phase.

We added text and further analysis for all spectrophotometry data for all variants at all the light intensities and the three incubation temperatures (30, 37 and 40 °C) to the manuscript and provided new data in Supplementary Table 13, Supplementary Table 14, Supplementary Figure 21, Supplementary Figure 22, and Supplementary Figure 23. We thank the reviewer for this suggestion, as this information might indeed be useful for readers.

- (2) In their rebuttal the authors reference some supplementary data from their prior paper (Baumschlager 2017). In my view, that paper has a wealth of additional results that support the premise of the current study, and it would be *very* helpful if they were explained in the introduction of this study. The 2017 paper investigates two classic VVD mutants, I85V and

I74V/I85V, which have well known effects on photocycling time, but observe that one of these mutants exhibits a massive change in light sensitivity (Fig 4 and S11), whereas the other does not. This shows that sensitivity and photocycle time can be tuned independently - they are not one-to-one - and that wild-type VVD is far from optimal in sensitivity. I agree with the authors' rebuttal that these results argue against sensitivity arising simply from variation in protein expression level. They also argue against the sensitivity being solely a function of lit-state lifetime.

The manuscript would be strengthened if the authors would describe something like this logic in their current manuscript, because it is absolutely not clear simply by reading this paper whether light sensitivity itself is an engineerable parameter, since so many other variables could in principle influence a dose response curve.

This prior work also suggests that the authors have already identified a very sensitive variant in I74V/I85V. Why was this variant not included in the current analyses, at least as a positive control? Would it be competitive with the best (most sensitive) variants discovered here?

Following the reviewer's suggestion, we added all relevant information to the introduction of the manuscript (pages 3-4). We thank the reviewer for this suggestion, as this additional text will improve the understanding of readers.

For clarification, we did not use I74/I85V as the variant would fail as a positive control in the current study because the increased sensitivity comes at the cost of a dramatically lower expression level. For this, compare the fluorescence level of strain AB363 that contains Opto-T7RNAP*(563) with nMagHigh1 and pMag, and strain AB364 which contains Opto-T7RNAP*(563-F2) that harbors the mutations I74V/I85V in Figure S20 (Baumschlager et al., 2017). The mCherry fluorescence values at 329 $\mu\text{W}/\text{cm}^2$, the maximal light intensity in this plot, were 35,459 a.u. for Opto-T7RNAP*(563) and 20,041 a.u. for Opto-T7RNAP*(563-F2). This means that this variant leads to a reduced reporter gene expression of 57% compared to the wild type. Even comparing the respective maximal expression of the two variants in all light conditions (20 $\mu\text{W}/\text{cm}^2$ for AB364 and 329 $\mu\text{W}/\text{cm}^2$ for AB363), the expression level in the I74V/I85V variant is reduced to 76% compared to the wild type. Since we did not want to compromise the activity of the Opto-T7RNAP for a higher light sensitivity, which supposedly would be easier to achieve, we focused on variants that show equal or higher expression levels compared to the wild-type. Thus, the variants identified in this study show equal or higher expression levels and increased light sensitivity.

(3) The authors' table detailing the sensitivity changes (Supp Table 13) is very useful. It is one of the key pieces of information for users who might be interested in implementing these mutations for further study. I would strongly prefer that this be included in the main text as a final figure panel or table, not on page 38 of the supplementary information where it is likely to escape notice by most readers.

The reviewer raises an important point. We have replaced the previous Figure 4 (the mammalian experiments) with this description, and moved the figure describing the mammalian experiment into the supplement according to the description in the next point.

(4) I have a major concern about the quantification and analysis of the mammalian cell data in Figure 3. It is quite misleading to only compare fold-changes between mutants in the light to the wild-type variant in the dark. This is because many variants already show dramatically increased gene expression in the dark! Fold-changes should be reported for each variant between its dark and light conditions. Separately, the authors can describe changes in "leakiness" in the dark, for example showing that M48V N100L is already ~10-fold higher than wild-type in the dark. It is not at all clear that the changes between dark and low-intensity light are significant for many variants (e.g. N100L or M48V/N100L). Is the fold-change between dark and 0.1 W/m² statistically different for the mutants compared to the wild-type in mammalian cells? Without such a comparison it is not clear that the changes in sensitivity extend to the mammalian context at all.

The reason for using fold-changes compared to the wild-type variant was to highlight the increase in overall expression level and not dark-to-light fold-changes of individual variants. However, fold changes in optogenetic experiments typically refer to dark-to-light-fold changes, as the reviewer correctly mentioned. We have chosen to show the absolute data (as fluorescence normalized to mRuby3 expression) in a newly created supplementary figure, which replaces previous Figure 4. Following the suggestion of the reviewer, we added the variant overview (previous Supp Table 13) as a new Figure 4 to bring further focus on the core aspects of this work, as the mammalian experiment is a mere extension of the findings to other organisms. This new supplementary figure clearly shows the difference in expression levels due to the different variants, which is the main value of these variants for mammalian systems. In addition, we mention in the text, that we observed increased expression levels, while fold-changes were not altered, as the reviewer pointed out.

We want to thank the reviewer again for their suggestions as we think that they helped us improve the manuscript.